# Locally private non-asymptotic testing of discrete distributions is faster using interactive mechanisms

**Berrett Thomas**
CREST, ENSAE, IP Paris
5, avenue Henry le Chatelier
91120 Palaiseau Cedex, FRANCE
`thomas.berrett@ensae.fr`

**Butucea Cristina**
CREST, ENSAE, IP Paris
5, avenue Henry le Chatelier
91120 Palaiseau Cedex, FRANCE
`cristina.butucea@ensae.fr`

## Abstract

We find separation rates for testing multinomial or more general discrete distributions under the constraint of $\alpha$-local differential privacy. We construct efficient randomized algorithms and test procedures, in both the case where only non-interactive privacy mechanisms are allowed and also in the case where all sequentially interactive privacy mechanisms are allowed. The separation rates are faster in the latter case. We prove general information theoretical bounds that allow us to establish the optimality of our algorithms among all pairs of privacy mechanisms and test procedures, in most usual cases. Considered examples include testing uniform, polynomially and exponentially decreasing distributions.

## 1   Introduction

Hypothesis testing of discrete distributions is intensively used as a first step in data based decision making and it is now also a component of many machine learning algorithms. Given samples from an unknown probability distribution $p$ and a known reference distribution $p_0$, the goal of a goodness-of-fit test is to decide whether $p$ fits $p_0$, or is signficantly different to it in some suitable sense. Here we will measure distance between distributions using either the $\mathbb{L}_1$ norm or the $\mathbb{L}_2$ norm, with our alternative hypotheses consisting of all distributions $p$ whose distance from $p_0$ is above a certain threshold. Our goal is to make accurate decisions, i.e. with low error probabilities, for distributions $p$ as close to $p_0$ as possible. The smallest separation between $p_0$ and the alternative hypothesis for which it remains possible to reliably distinguish between the two hypotheses is known as the uniform separation rate, $\delta$. Its optimality is proven by showing that, whenever $p$ is closer to $p_0$ than $\delta$, no test procedure will be able to distinguish them with small error probabilities. As shown by Valiant and Valiant [2014, 2017], the dependence of $\delta$ on $p_0$ in the standard problem without privacy constraints is pronounced and intricate.

In this work, we quantify how the constraint of local differential privacy affects the optimal separation rate. Differential privacy Dwork et al. [2006] is the most popular formalism under which the analyst statistically randomizes data to be published in order to protect the privacy of the individuals in the study. The way in which the data is randomized, known as the privacy mechanism, must be carefully chosen to preserve the information in the data that is most pertinent for the task at hand, though it is well-established that the cost of protecting privacy is necessarily a deterioration in statistical performance. Local differential privacy, in which there is no trusted curator who has access to all the original data, is more stringent than the original differential privacy constraint, and it is often observed in the literature that in this context we experience a further deterioration of the achievable performance of estimation and test procedures, which are allowed to use private data only.

Local differential privacy can be attained through several kinds of privacy mechanisms. They can be non-interactive (also known as private-coin) when the users independently randomize the sample

they receive, or interactive in the sense that some information is shared. We consider a large class of sequentially interactive mechanisms where some information (e.g. the previously privatized sample) can be transmitted from one user to the next. In this setup, the set of public-coin mechanisms, such as those considered in Acharya et al. [2019c], is a subset of all sequentially interactive mechanisms where the shared information is the original seed the first user employed.

## 1.1 Our contributions

We find the optimal separation rate around an arbitrary discrete distribution $p_0$ on $\mathbb{N}$ under local differential privacy constraints and this optimality involves two steps. On the one hand, we provide efficient and statistically optimal pairs of privacy mechanisms and their associated test procedures whose error probabilities are small for all distributions $p$ further from $p_0$ than $\delta$. On the other hand, we show that whenever $p$ is closer to $p_0$ than the separation rate $\delta$, no pair of privacy mechanism and test procedure is able to distinguish $p$ and $p_0$. We show that faster rates are attainable using interactive privacy mechanisms rather than non-interactive. As indicated below, previous works have found optimal rates in the special case of $p_0$ being a uniform distribution, and upper bounds for general finite supported $p_0$, in the $\mathbb{L}_1$ problem. Here, we provide optimal rates for general $p_0$ supported on $\mathbb{N}$ that can be quicker than in the uniform case (see Table 1), we treat both the $\mathbb{L}_1$ and the $\mathbb{L}_2$ test problems in parallel, we introduce new privacy mechanisms and test procedures, and we provide shorter proofs and more explicit upper and lower bounds. The interactive mechanism that we use is a two-step procedure: for the first half of the sample we employ a Laplace mechanism and estimate the unknown probabilities, for the second half we randomize encoding this information and build a $\ell_2$-type statistic. Our optimality results show that the separation rates are optimal in most usual cases. Let us stress the fact that the second part of Theorem 5 is particularly useful in the case of noninteractive mechanisms where, as it was also noted by Lam-Weil et al. [2020], the usual inequalities in Duchi et al. [2018] can only result in suboptimal lower bounds. We highlight the examples of (nearly) uniform distributions, and distributions with polynomially or exponentially decreasing tails. All results are valid non-asymptotically, that is with a finite number of samples.

## 1.2 Related work

The study of discrete data under local privacy constraints can be traced back at least as far as Warner [1965], in which the classical randomised response mechanism was introduced to provide privacy when estimating the proportion of a population that belongs to a certain group. This problem can be thought of as a special case, with an alphabet of size two, of the problem of estimating the probability vector of a multinomial random variable. For a general (finite) alphabet size, Duchi et al. [2013] derive upper and lower bounds on the minimax estimation risk in this more general problem for both the $\mathbb{L}_1$ and $\mathbb{L}_2$ metrics, and, in particular, show that a generalization of the randomized response algorithm is rate-optimal in certain regimes. Besides the standard estimation problem, one can also consider the problem of estimating succinct histograms, or heavy hitters; see, for example, Bassily and Smith [2015].

Compared with estimation problems, hypothesis testing is relatively under-explored in the setting of local differential privacy. Early work includes Kairouz et al. [2014, 2016], in which the aim is to test the simple hypotheses $H_0 : P = P_0$ vs. $H_1 : P = P_1$, where $P_0$ and $P_1$ are two given discrete distributions. Under a mutual information-based local privacy constraint, Liao et al. [2017] considered the more general problem of $m$-ary hypothesis testing. The goodness-of-fit testing problem, which we consider in this paper, was investigated in Gaboardi and Rogers [2017], Sheffet [2018], where the authors provide analyses of procedures based on chi-squared tests and the optimal non-private test of Valiant and Valiant [2014, 2017], respectively, under specific privacy mechanisms. Unfortunately, these privacy mechanisms and tests are typically suboptimal, attaining slower separation rates. As we show here, a corrected chi-squared statistic, calculated using suitably generated private data, may be optimal for testing uniformity, but for general null hypotheses a more subtle procedure is required. In $\mathbb{L}_1$, or $TV$, distance Acharya et al. [2019a] studies the problem of testing uniformity, and provides tests that they show are optimal among all tests using their chosen privacy mechanisms. Techniques for proving lower bounds over general classes of privacy mechanisms are developed in Acharya et al. [2019b] and applied to uniformity testing. Upper bounds for more general $p_0$ in the $\mathbb{L}_1$ setting (in a different form to our rates) can be found in Acharya et al. [2019c, Appendix D]. These more general upper bounds follow from using a technique, first introduced by Goldreich [2016], whereby the problem of testing with null $p_0$ can be reduced to uniformity testing; however,

the optimality of this approach is not proved and lower bounds do not follow. An indirect proof of the interactive lower bound in the $\mathbb{L}_1$ case with $p_0$ uniform, can be found in Amin et al. [2019]. The role of interactivity in locally private testing is studied in Joseph et al. [2019], where it is shown that an optimal procedure for testing the hypotheses $H_i : P \in \mathcal{P}_i, i = 1, 2$, for disjoint, convex $\mathcal{P}_i$, is non-interactive. This is in contrast to our results, which show that, in goodness-of-fit testing with a general $p_0$, interactive procedures achieve significantly faster separation rates.

In the non-private setting, goodness-of-fit testing of discrete distributions has recently received a great deal of attention. Valiant and Valiant [2014, 2017] found near optimal separation rates that show that the difficulty of the problem depends intricately on the specific null hypothesis; see also Diakonikolas and Kane [2016] and the survey article Balakrishnan and Wasserman [2018]. The gap between the upper and lower bounds in this problem has been further explored in Blais, Canonne and Gur [2019]. The problem has also been considered in the non-local differentially private setting [Wang et al., 2015, Gaboardi et al., 2016, Cai et al., 2017, Aliakbarpour et al., 2018, Acharya et al., 2018]. Upper bounds have been provided, which have been shown to be nearly optimal in certain regimes for the case of a uniform null. Besides goodness-of-fit, there is also work in this setting on testing independence between discrete variables [Wang et al., 2015, Gaboardi et al., 2016], and in testing independence between a discrete and a continuous variable by constructing differentially private versions of classical rank-based tests [Couch et al., 2019].

## 2    Preliminaries

Let $\mathcal{P}_d = \{p = (p(1), \ldots, p(d)) \in [0,1]^d : \sum_{j=1}^d p(j) = 1\}$ denote the set of all probability vectors in $d$-dimensions. For $x = (x(1), \ldots, x(d)) \in \mathbb{R}^d$ write $\|x\|_1 = \sum_{j=1}^d |x(j)|$ for the $\ell_1$-norm and $\|x\|_2$ for the Euclidean norm of $x$. For $p \in \mathcal{P}_d$ we say that $X$ is distributed according the probability $p$, $X \sim p$, if $\mathbb{P}(X = j) = p(j)$ for each $j = 1, \ldots, d$. Given $p_0 \in \mathcal{P}_d, \delta > 0$ and data $X_1, \ldots, X_n \overset{\text{i.i.d}}{\sim} p$ we will study the $\mathbb{L}_1$ and $\mathbb{L}_2$ problems of testing the hypotheses:

$$H_0 : p = p_0 \quad \text{vs.} \quad H_1(\delta, \mathbb{L}_r) : \{p \in \mathcal{P}_d \text{ such that } \|p - p_0\|_r \geq \delta\}, \tag{1}$$

for $r$ equal to 1 and 2 respectively, under an $\alpha$-local differential privacy (LDP) constraint on the allowable tests. An $\alpha$-LDP privacy mechanism $Q$ generates private data $Z_i$ taking values in $\mathcal{Z}$ via the conditional distribution $Q_i(\cdot|x_i, z_1, ..., z_{i-1})$ such that

$$\sup_A \sup_{z_1, ..., z_{i-1}} \sup_{x,x'} \frac{Q_i(A|x, z_1, ..., z_{i-1})}{Q_i(A|x', z_1, ..., z_{i-1})} \leq e^\alpha, \text{ for all } i = 1, ..., n.$$

Given an $\alpha$-LDP privacy mechanism $Q$, let $\Phi_Q = \{\phi : \mathcal{Z}^n \to [0,1]\}$ denote the set of all (randomized) tests of the hypotheses (1) based on $Z_1, \ldots, Z_n$. We can define the minimax testing risk when using the privacy mechanism $Q$ by

$$\mathcal{R}_n(p_0, \delta, Q, \mathbb{L}_r) := \inf_{\phi \in \Phi_Q} \sup_{p \in H_1(\delta, \mathbb{L}_r)} \left\{ \mathbb{E}_{p_0}(\phi) + \mathbb{E}_p(1 - \phi) \right\}.$$

Then, writing $\mathcal{Q}_\alpha$ for the collection of all $\alpha$-LDP sequentially-interactive privacy mechanisms, we can define the $\alpha$-LDP minimax testing risk of (1) by

$$\mathcal{R}_{n,\alpha}(p_0, \delta, \mathbb{L}_r) := \inf_{Q \in \mathcal{Q}_\alpha} \mathcal{R}_n(p_0, \delta, Q, \mathbb{L}_r).$$

Given $\gamma \in (0, 1)$ we aim to find the $\alpha$-LDP minimax testing radius defined by

$$\mathcal{E}_{n,\alpha}(p_0, \mathbb{L}_r) := \inf\{\delta > 0 : \mathcal{R}_{n,\alpha}(p_0, \delta, \mathbb{L}_r) \leq \gamma\}$$

Moreover, we will also aim to find the minimax testing risk of (1) under the additional restriction that $Q$ is a non-interactive $\alpha$-LDP privacy mechanism. Letting $\mathcal{Q}_\alpha^{\text{NI}} \subset \mathcal{Q}_\alpha$ denote the subset of all such privacy mechanisms, we similary define

$$\mathcal{R}_{n,\alpha}^{\text{NI}}(p_0, \delta, \mathbb{L}_r) := \inf_{Q \in \mathcal{Q}_\alpha^{\text{NI}}} \mathcal{R}_n(p_0, \delta, Q, \mathbb{L}_r), \quad \mathcal{E}_{n,\alpha}^{\text{NI}}(p_0, \mathbb{L}_r) := \inf\{\delta > 0 : \mathcal{R}_{n,\alpha}^{\text{NI}}(p_0, \delta, \mathbb{L}_r) \leq \gamma\}.$$

Note that this formalism is equivalent to finding the smallest sample size $n$ in order to attain a given accuracy, i.e. testing risk measure.

# 3 Building private samples and optimal test procedures

We split the support of the multinomial distribution $p_0$ into a main set $B$ and a tail set $B^c$. As is now typical in such problems since Valiant and Valiant [2014], we combine a $\ell_2$ test on $B$ (which is not the usual bias-corrected $\ell_2$ statistic) with a tail-test on $B^c$ in order to achieve optimality.

These tests procedures use privatized data. Our non-interactive privacy mechanisms use classical Laplace randomization – see, for example, Duchi et al. [2018]. The interactive procedure involved in the $\chi^2$-test is novel in the context of discrete distributions; see Butucea et al. [2020] for a similar mechanism in the continuous setting. It is a two-step procedure, that uses part of the sample in order to estimate the frequencies $\widehat{p}_j$ and then randomizes the other part of the sample using a censored value of $\widehat{p}_j - p_0(j)$. A simple average of this second part of the private sample allows us to construct the $\ell_2$-test. Thus the latter procedure encodes partial information on the distribution in the randomization of the second part of the sample and benefits from it.

## 3.1 Non-interactive privacy mechanisms

Assume that the sample size is even and that the data is given by $X_1, \ldots, X_{2n}$. Given a nonempty subset $B \subseteq [d]$, which will typically contain the bulk of the distribution, we define a non-interactive privacy mechanism $Q_B \in \mathcal{Q}_\alpha^{\mathrm{NI}}$ and a test $\phi_B \in \Phi_{Q_B}$ as follows. Given an i.i.d. sequence $(W_{ij})_{i \in [n], j \in B}$ of Laplace(1) random variables, for each $i \in [n]$ and $j \in B$ write

$$Z_{ij} := \mathbb{1}_{\{X_i = j\}} + \frac{2}{\alpha} W_{ij}. \tag{2}$$

We have that $(Z_{ij})_{i \in [n], j \in B}$ is an $\alpha$-LDP version of $X_1, \ldots, X_n$ [see, e.g., Gaboardi and Rogers, 2017]. We will estimate the quantity $\sum_{j \in B} (p - p_0)^2(j)$ using the $\ell_2$ statistic

$$S_B := \sum_{j \in B} \frac{1}{n(n-1)} \sum_{i_1 \neq i_2} \{Z_{i_1 j} - p_0(j)\}\{Z_{i_2 j} - p_0(j)\}.$$

Letting $(W_i)_{i=n+1}^{2n}$ denote a second sequence of i.i.d. Laplace(1) random variables, for $i = n+1, \ldots, 2n$ we set

$$Z_i = \mathbb{1}_{\{X_i \in B^c\}} + \frac{2}{\alpha} W_i.$$

Then again $(Z_i)_{i=n+1}^{2n}$ is an $\alpha$-LDP version of $X_{n+1}, \ldots, X_{2n}$ version of $X_{n+1}, \ldots, X_{2n}$. Further, define

$$T_B := \frac{1}{n} \sum_{i=n+1}^{2n} \{Z_i - p_0(B^c)\},$$

for the tail test statistic, where we write $p_0(B^c) = \sum_{j \in B^c} p_0(j)$. Note that $T_B$ is an unbiased estimator of $(P - P_0)(B^c)$. With the critical values $C_{1,B} := \{656|B|/(n(n-1)\alpha^4\gamma)\}^{1/2}$ and $C_{2,B} := 6/(n\alpha^2\gamma)^{1/2}$, we finally set

$$\phi_B(Z_1, \ldots, Z_{2n}) := \begin{cases} 1 & \text{if } S_B \geq C_{1,B} \text{ or } T_B \geq C_{2,B} \\ 0 & \text{otherwise} \end{cases},$$

that is, we reject $H_0$ if either the $\ell_2$ test rejects on the bulk $B$, i.e. $S_B \geq C_{1,B}$, or $T_B \geq C_{2,B}$ and the tail-test rejects.

**Theorem 1.** *When $\alpha \in (0, 1]$, for any $\emptyset \neq B \subseteq [d]$ we have that*

$$\mathcal{E}_{n,\alpha}^{\mathrm{NI}}(p_0, \mathbb{L}_1) \leq 8 \max\left[12\left\{\frac{|B|^3}{n(n-1)\alpha^4\gamma^2}\right\}^{1/4}, p_0(B^c)\right].$$

*and*

$$\mathcal{E}_{n,\alpha}^{\mathrm{NI}}(p_0, \mathbb{L}_2) \leq 8 \max\left[12\left\{\frac{|B|}{n(n-1)\alpha^4\gamma^2}\right\}^{1/4}, p_0(B^c)\right].$$

Note that we can actually include discrete distributions on all of $\mathbb{N}$. We prove the tightest upper bounds by finding the sets $B$ that minimize the right-hand sides in Theorem 1. The search algorithm

is trivial if we order the sequence $p_0(\cdot)$ in decreasing order. Indeed, then it is straightforward to see that the optimal $B$ is of the form $\{1, \ldots, j\}$ in both cases, with the first term in the maximum increasing with $j$, and $p_0(B^C)$ decreasing with $j$. Therefore, there are always finite sets (possibly large) that minimize the right-hand sides.

Theorem 1 yields the following immediate corollary.

**Corollary 2.** *Let*

$$j_* = j_*(n\alpha^2, p_0, \mathbb{L}_1) := \min\left\{ j = 1, \ldots, d : \frac{j^{3/4}}{(n\alpha^2)^{1/2}} \geq \sum_{j'=j+1}^{d} p_0(j') \right\}$$

$$j_{**} = j_{**}(n\alpha^2, p_0, \mathbb{L}_2) := \min\left\{ j = 1, \ldots, d : \frac{j^{1/4}}{(n\alpha^2)^{1/2}} \geq \sum_{j'=j+1}^{d} p_0(j') \right\}.$$

*When $\alpha \in (0, 1]$, there exist $C_1 = C_1(\gamma)$ and $C_2 = C_2(\gamma)$ such that*

$$\mathcal{E}_{n,\alpha}^{\mathrm{NI}}(p_0, \mathbb{L}_1) \leq C_1 \frac{j_*^{3/4}}{(n\alpha^2)^{1/2}} \quad and \quad \mathcal{E}_{n,\alpha}^{\mathrm{NI}}(p_0, \mathbb{L}_2) \leq C_2 \frac{j_{**}^{1/4}}{(n\alpha^2)^{1/2}}.$$

In particular, for testing the uniform distribution over $[d]$, this corollary shows in both cases a loss of a factor $d^{1/4}$ with respect to the minimax rates that we can attain without privacy.

In Corollary 2 we always have $j_*, j_{**} \leq d$, so we can always say that $\mathcal{E}_{n,\alpha}^{\mathrm{NI}}(p_0, \mathbb{L}_1) \lesssim d^{3/4}/(n\alpha^2)^{1/2}$ and that $\mathcal{E}_{n,\alpha}^{\mathrm{NI}}(p_0, \mathbb{L}_2) \lesssim d^{1/4}/(n\alpha^2)^{1/2}$. However, for some values of $p_0$ our upper bound is better than this.

It is important to note here that the $\mathbb{L}_1$ test behaves very differently in this context from the case of non private setup. It has been known since Valiant and Valiant [2014], see also Balakrishnan and Wasserman [2018], that in the direct setup a weighted $\ell_2$-test is needed in order to attain the optimal rates. This is due to the heteroscedasticity of the multinomial model (the variances of the counts are proportional to their probabilities) and a correction for very small variances needs to be included. Unlike this setup, the privacy constraint induces an unavoidable homoscedastic term in the variance of the $\ell_2$ test and makes the correction useless in this case, resulting in a loss in the rate. We will see in Section 4 that these rates for the $\mathbb{L}_1$ problem are essentially optimal.

The $\mathbb{L}_2$ test also combines the $\ell_2$ and the tail tests in order to achieve nearly optimal rates and this is also in contrast with the non-private case where the $\ell_2$ test is sufficient. However, as we will describe in Section 5, for polynomially decreasing distributions there is a gap between our non-interactive upper and lower bounds in some cases. Nevertheless, our results in Sections 3.2 and 4 do demonstrate a significant gap between non-interactive and interactive rates, even in these settings.

### 3.2  Interactive privacy mechanisms and faster rates

Assume here that the sample is split in 3 parts, or that the data is given by $X_1, \ldots, X_{3n}$. The data $X_1, ..., X_{2n}$ is used to build the interactive test statistic $D_B$ as described hereafter, while the third part of the sample, $X_{2n+1}, ..., X_{3n}$, is used to build the same test statistic $T_B$ as in the noninteractive setup.

The main difference between our interactive and non-interactive procedures is in how we estimate the $\ell_2$-type statistic on the bulk $B$. We use the first half of the observations to gain partial information on probabilities $p_j$ with $j$ in the bulk $B$, and then use these estimated values in the randomization of the second half of the sample. This 2-step procedure is interactive but not of the public-coin type. For simplicity of exposition, we in fact estimate this $\ell_2$-type statistic over the whole alphabet.

We define an interactive privacy mechanism $Q_{\mathrm{I}} \in \mathcal{Q}_\alpha$ and a test $\psi_B \in \Phi_{Q_{\mathrm{I}}}$ as follows. With the first half of the sample, as in (2) with $B = [d]$, generate an i.i.d. sequence $(W_{ij})_{i \in [n], j \in [d]}$ of Laplace(1) random variables, and for each $i \in [n]$ and $j \in [d]$ write

$$Z_{ij} := \mathbb{1}_{\{X_i = j\}} + \frac{2}{\alpha} W_{ij}.$$

We again have that $(Z_{ij})_{i\in[n],j\in[d]}$ is an $\alpha$-LDP version of $X_1,\ldots,X_n$. For each $j \in [d]$ set

$$\widehat{p}_j := \frac{1}{n}\sum_{i=1}^{n} Z_{ij}.$$

Set $c_\alpha = \frac{e^\alpha+1}{e^\alpha-1}$ and $\tau = (n\alpha^2)^{-1/2}$. As for the second half of the sample, for each $i = n+1,\ldots,2n$, generate $Z_i$ taking values in $\{-c_\alpha\cdot\tau, c_\alpha\cdot\tau\}$ with probabilities depending on the previously privatized samples through $\widehat{p}_j$ :

$$\mathbb{P}(Z_i = c_\alpha \cdot \tau | X_i = j) = \frac{1}{2}\Big(1 + \frac{[\widehat{p}_j - p_0(j)]^\tau_{-\tau}}{c_\alpha\cdot\tau}\Big),$$

where we denote by

$$[v]^\tau_{-\tau} = (-\tau) \vee v \wedge \tau, \quad \text{for all } v \in \mathbb{R}$$

the censoring operator. Then $(Z_i)_{i=n+1,\ldots,2n}$ is an $\alpha$-LDP version of $X_{n+1},\ldots,X_{2n}$ [Butucea et al., 2020]. We then define the test statistic

$$D_B = \frac{1}{n}\sum_{i=n+1}^{2n} Z_i - \sum_{j=1}^{d} p_0(j)\{[\widehat{p}_j - p_0(j)]^\tau_{-\tau}\}$$

and $C_3 := \frac{e+1}{e-1}\frac{(4/\gamma)^{1/2}}{n\alpha^2}$. The final test is

$$\psi_B(Z_1,\ldots,Z_{3n}) := \left\{ \begin{array}{ll} 1 & \text{if } D_B \geq C_3 \text{ or } T_B \geq C_{2,B} \\ 0 & \text{otherwise} \end{array} \right.,$$

that is, we reject $H_0$ if either $D_B \geq C_3$ or $T_B \geq C_{2,B}$.

**Theorem 3.** *There exists a universal constant $C$ such that when $\alpha \in (0,1]$, for any $\emptyset \neq B \subseteq [d]$, we have that*

$$\mathcal{E}_{n,\alpha}(p_0,\mathbb{L}_1) \leq C \max\Big\{\frac{|B|^{1/2}}{(n\alpha^2\gamma^2)^{1/2}}, p_0(B^c)\Big\} \quad \text{and} \quad \mathcal{E}_{n,\alpha}(p_0,\mathbb{L}_2) \leq \frac{C}{(n\alpha^2\gamma^2)^{1/2}}.$$

*In particular, let*

$$\tilde{j} = \tilde{j}(n\alpha^2, p_0, \mathbb{L}_1) := \min\Big\{j = 1,\ldots,d : \frac{j^{1/2}}{(n\alpha^2)^{1/2}} \geq \sum_{j'=j+1}^{d} p_0(j')\Big\}.$$

*Then there exists $C_1 = C_1(\gamma)$ such that, when $\alpha \in (0,1]$, we have*

$$\mathcal{E}_{n,\alpha}(p_0,\mathbb{L}_1) \leq C_1 \frac{\tilde{j}^{1/2}}{(n\alpha^2)^{1/2}}.$$

## 4   Non-asymptotic optimality

Attaining the rates through a particular randomization of the original sample and an associated test scheme does not prevent us from trying to improve on these choices. Instead, our lower bound results show that there are no better choices of privacy mechanisms and test procedures that would improve the test risk (or the separation rate) uniformly over the set of discrete distributions. It is of particular interest to show that no other $\alpha$-LDP Markov kernels could be combined with any of the tests to improve on the upper bounds of our rates. There are however multiple choices of such couples leading to the optimal rates that we have described.

Proving the optimality of our methods consists of building a family $\{p_\xi : \xi \in \mathcal{V}\}$ that belongs to the alternative set of probability distributions $H_1(\delta)$ with high probability and then reducing the test problem to testing between $p_0$ under the null and the mixture of the $p_\xi$ under the alternative.

**Proposition 4.** *If $\{p_\xi : \xi \in \mathcal{V}\}$ is a family of distributions such that*

$$P_\xi(p_\xi \notin H_1(\delta)) \leq \gamma_1, \quad \text{for some } \gamma_1 > 0.$$

*then, for arbitrary $\eta$ in (0,1), we have*

$$\mathcal{R}_{n,\alpha}(p_0,\delta) \geq \inf_{Q\in\mathcal{Q}_\alpha}(1-\eta)\Big(1 - \frac{1}{\eta}TV(QP_0^n, E_\xi QP_\xi^n)\Big) - \gamma_1.$$

It is sufficient to show that $TV(QP_0^n, E_\xi QP_\xi^n) \leq \eta \cdot \gamma_2$ such that $(1-\eta)(1-\gamma_2) - \gamma_1 \geq \gamma$. Standard inequalities prove that it is sufficient to bound from above the Kullback–Leibler or the $\chi^2$ discrepancy between the private distribution under the null and the average of conveniently chosen private distributions under the set of alternatives.

The way the previous discrepancies relate to the underlying distributions of the data proves to be significantly different in the cases when we are constrained to use non-interactive privacy mechanisms only, and when we are allowed to use any privacy mechanism.

**Information theoretical bounds for testing.** For maximal generality, we assume that the privacy mechanisms may act differently on each sample $X_i$. An interactive procedure acts through $q_i(z_i|X_i = j, z_1, ..., z_{i-1})$ on $X_i$ and the resulting $Z_i$ is distributed, conditionally on $Z_1, ..., Z_{i-1}$, according to $m_i^\xi(z_i|Z_1, ..., Z_{i-1}) = p_\xi^\top q_i(z_i|\cdot, Z_1, ..., Z_{i-1})$. A non-interactive procedure acts simply through $q_i(z_i|X_i = j)$ on $X_i$ and the resulting $Z_i$ is distributed according to $m_i^\xi(z_i) = p_\xi^T q_i(z_i|\cdot)$.

**Theorem 5.** *Given the previous family of distributions $\{p_\xi : \xi \in \mathcal{V}\}$, we have*

$$KL(QP_0^n, E_\xi QP_\xi^n) \leq E_\xi \left[ (p_\xi - p_0)^\top \Omega(p_\xi - p_0) \right],$$

*where the matrix $\Omega$ has elements*

$$\Omega_{j,k} = \sum_{i=1}^n \mathbb{E}_{p_0} \int \left( \frac{q_i(z_i|j, Z_1, ..., Z_{i-1})}{m_i^0(z_i|Z_1, ..., Z_{i-1})} - 1 \right) \left( \frac{q_i(z_i|k, Z_1, ..., Z_{i-1})}{m_i^0(z_i|Z_1, ..., Z_{i-1})} - 1 \right) m_i^0(z_i|Z_1, ..., Z_{i-1}) dz_i.$$

*In the particular case of non-interactive privacy mechanisms, we have for independent copies $\xi, \xi'$*

$$\chi^2(QP_0^n, E_\xi QP_\xi^n) \leq E_{\xi,\xi'} \left[ \exp \left( (p_\xi - p_0)^\top \Omega(p_{\xi'} - p_0) \right) \right] - 1,$$

*where $\Omega$ takes the simpler form*

$$\Omega_{j,k} = \sum_{i=1}^n \int \left( \frac{q_i(z_i|j)}{m_i^0(z_i)} - 1 \right) \left( \frac{q_i(z_i|k)}{m_i^0(z_i)} - 1 \right) m_i^0(z_i) dz_i.$$

## 4.1 Non-interactive approach

Recall that $\mathcal{E}_{n,\alpha}^{\mathrm{NI}}(p_0)$ is the $\alpha$-LDP minimax testing radius when we restrict to non-interactive privacy mechanisms. We have the following result.

**Theorem 6.** *There exist $c_1 = c_1(\gamma) > 0$ and $c_2 = c_2(\gamma) > 0$ such that for all $\alpha \in (0, 1]$ we have*

$$\mathcal{E}_{n,\alpha}^{\mathrm{NI}}(p_0, \mathbb{L}_1) \geq c_1 \max_{j=1,...,d} \min \left\{ \frac{j^{3/4}}{(n\alpha^2)^{1/2}}, \frac{jp_0(j)}{\log^{1/2}(2j)} \right\}$$

*and*

$$\mathcal{E}_{n,\alpha}^{\mathrm{NI}}(p_0, \mathbb{L}_2) \geq c_2 \max_{j=1,...,d} \min \left\{ \frac{j^{1/4}}{(n\alpha^2)^{1/2}}, \frac{j^{1/2}p_0(j)}{\log^{1/2}(2j)} \right\}.$$

We have the following immediate corollary.

**Corollary 7.** *Let*

$$\ell_* = \ell_*(n\alpha^2, p_0, \mathbb{L}_1) := \max \left\{ j = 1, \ldots, d : \frac{j^{3/4}}{(n\alpha^2)^{1/2}} \leq \frac{jp_0(j)}{\log^{1/2}(2j)} \right\}$$

$$\ell_{**} = \ell_{**}(n\alpha^2, p_0, \mathbb{L}_2) := \max \left\{ j = 1, \ldots, d : \frac{j^{1/4}}{(n\alpha^2)^{1/2}} \leq \frac{j^{1/2}p_0(j)}{\log^{1/2}(2j)} \right\}.$$

*Then there exist $c_1 = c_1(\gamma) > 0$ and $c_2 = c_2(\gamma) > 0$ such that when $\alpha \in (0, 1]$ we have*

$$\mathcal{E}_{n,\alpha}^{\mathrm{NI}}(p_0, \mathbb{L}_1) \geq c_1 \frac{\ell_*^{3/4}}{(n\alpha^2)^{1/2}} \quad \text{and} \quad \mathcal{E}_{n,\alpha}^{\mathrm{NI}}(p_0, \mathbb{L}_2) \geq c_2 \frac{\ell_{**}^{1/4}}{(n\alpha^2)^{1/2}}.$$

According to the behaviour of $p_0$, we may have identical or different values for $\ell_*$ and $\ell_{**}$.

In many examples of interest, these lower bounds match our previous upper bounds in Corollary 2 up to log factor, even though $\ell_*$ and $\ell_{**}$ do not solve exactly the same problems as $j_*$ and $j_{**}$, respectively.

## 4.2 Interactive approach

Under their most general form the privacy mechanisms we allow are sequentially interactive. As shown by Theorem 3 and Corollary 7, the optimal rates for testing are faster with interactive procedures than with non-interactive procedures. The following theorem shows that the upper bounds in Theorem 3 are optimal.

**Theorem 8.** *There exist $c_1 = c_1(\gamma) > 0$ and $c_2 = c_2(\gamma) > 0$ such that when $\alpha \in (0,1]$ we have*

$$\mathcal{E}_{n,\alpha}(p_0, \mathbb{L}_1) \geq c_1 \max_{j=1,\ldots,d} \min\left\{ \frac{j^{1/2}}{(n\alpha^2)^{1/2}}, \frac{p_0(j)}{\log^{1/2}(2j)} \right\}$$

*and*

$$\mathcal{E}_{n,\alpha}(p_0, \mathbb{L}_2) \geq c_2 \frac{1}{(n\alpha^2)^{1/2}}.$$

*In particular, let*

$$\tilde{\ell} = \tilde{\ell}(n\alpha^2, p_0, \mathbb{L}_1) := \max\left\{ j = 1,\ldots,d : \frac{j^{1/2}}{(n\alpha^2)^{1/2}} \leq \frac{p_0(j)}{\log^{1/2}(2j)} \right\}.$$

*Then there exist $c_1 = c_1(\gamma) > 0$ such that when $\alpha \in (0,1]$ we have*

$$\mathcal{E}_{n,\alpha}(p_0, \mathbb{L}_1) \geq c_1 \frac{\tilde{\ell}^{1/2}}{(n\alpha^2)^{1/2}}.$$

## 5 Particular classes of distributions

In this section we explicitly calculate the separation rates in several examples. See Section 5 in the supplementary material for more detailed and more general calculations. Inequalities $\gtrsim$ are valid up to $\log$ factors.

**Nearly uniform distributions** Suppose that $p_0(j) \propto j^{-\beta}$ for some $\beta \in [0,1)$, and that $d^{3/4}/(n\alpha^2)^{1/2} \leq (1-\beta)/(\log^{1/2}(2d))$. Then

$$\frac{dp_0(d)}{\log^{1/2}(2d)} = \frac{d^{1-\beta}}{\log^{1/2}(2d)\sum_{\ell=1}^{d}\ell^{-\beta}} \geq \frac{d^{1-\beta}}{\log^{1/2}(2d)\int_0^d x^{-\beta}\,dx} = \frac{1-\beta}{\log^{1/2}(2d)} \geq \frac{d^{3/4}}{(n\alpha^2)^{1/2}},$$

and it follows that $\ell_* = d$. Thus, in this setting, $\mathcal{E}_{n,\alpha}^{\mathrm{NI}}(p_0, \mathbb{L}_1) \gtrsim d^{3/4}/(n\alpha^2)^{1/2}$. Concerning the $\mathbb{L}_2$ rates, $\mathcal{E}_{n,\alpha}^{\mathrm{NI}}(p_0, \mathbb{L}_2) \gtrsim d^{1/4}/(n\alpha^2)^{1/2}$ if $\beta \leq 1/4$, whereas it is $\gtrsim d^{1/4}/(n\alpha^2)^{1/2} \wedge d^{1/2-\beta} \wedge (n\alpha^2)^{-(\beta-1/2)/(2\beta-1/2)}$ if $\beta > 1/4$.

**Polynomially decreasing distributions** Suppose that $p_0(j) \propto j^{-1-\beta}$ for some $\beta > 0$, as for example is the case for the Pareto distributions used in extreme value theory. It is shown in the Supplementary material that, when $1 \leq n\alpha^2 \leq (d/C)^{2\beta+3/2}$ we have that $j_* \leq \lceil C(n\alpha^2)^{1/(2\beta+3/2)} \rceil$. On the other hand, if $n\alpha^2 > (d/C)^{2\beta+3/2}$ then we will just say that $j_* \leq d$. It follows that

$$\mathcal{E}_{n,\alpha}^{\mathrm{NI}}(p_0, \mathbb{L}_1) \lesssim \frac{j_*^{3/4}}{(n\alpha^2)^{1/2}} \lesssim \min\left\{ (n\alpha^2)^{-\frac{2\beta}{4\beta+3}}, \frac{d^{3/4}}{(n\alpha^2)^{1/2}} \right\}.$$

From the corresponding lower bounds, we get

$$\mathcal{E}_{n,\alpha}^{\mathrm{NI}}(p_0, \mathbb{L}_1) \gtrsim \left\{ n\alpha^2 \log^{3/(4\beta)}(n\alpha^2) \right\}^{-2\beta/(4\beta+3)} \wedge \frac{d^{3/4}}{(n\alpha^2)^{1/2}}.$$

So, the lower bounds match the upper bounds up to $\log$ factors in this case.

**Exponentially decreasing distributions** Suppose that $p_0(j) \propto \exp(-j^\beta)$ for some $\beta > 0$. More generally, we may include the geometric distribution with $p_0(j) \propto p^j = \exp(-j\log(1/p))$ or $p_0(j) \propto j^\eta \exp(-cj^\beta)$, for $\eta$ real number and $c, \beta > 0$. The upper bounds match the lower bounds in this case, and lead e.g. in the case of noninteractive privacy mechanisms and $\mathbb{L}_1$ norm to the rate

$$\mathcal{E}_{n,\alpha}^{\mathrm{NI}}(p_0, \mathbb{L}_1) \asymp \min\left\{ \frac{\log^{3/(4\beta)}(n\alpha^2)}{\sqrt{n\alpha^2}}, \frac{d^{3/4}}{\sqrt{n\alpha^2}} \right\}.$$

Analogous calculations can be done for interactive mechanisms and $\mathbb{L}_2$ norm. Table 1 summarizes the minimax separation rates, for examples of distrbution probabilities $p_0$. They are optimal up to $\log$ factors except for the $\mathbb{L}_2$ distance in the case of uniform and polynomially decreasing distributions.

Table 1: Separation rates for testing discrete distributions. The non-interactive $\mathbb{L}_1$ rate in the uniform example, marked by ($\dagger$), was previously established by Acharya et al. [2019a] and Acharya et al. [2019b]. An upper bound for interactive procedures, equal to the optimal rate established in this work and marked here by ($*$), was also established by Acharya et al. [2019a].

| $p_0$ | Noninteractive | | Interactive | |
|---|---|---|---|---|
| | $\mathbb{L}_1$ | $\mathbb{L}_2$ | $\mathbb{L}_1$ | $\mathbb{L}_2$ |
| Uniform$[d]$ | $\frac{d^{3/4}}{\sqrt{n\alpha^2}}$ ($\dagger$) | $\leq \frac{d^{1/4}}{\sqrt{n\alpha^2}}$ $\gtrsim \frac{d^{1/4}}{\sqrt{n\alpha^2}} \wedge \frac{1}{\sqrt{d}}$ | $\frac{d^{1/2}}{\sqrt{n\alpha^2}}$ ($*$) | $\frac{1}{\sqrt{n\alpha^2}}$ |
| $\propto j^{-1-\beta}$ | $(n\alpha^2)^{-\frac{2\beta}{4\beta+3}} \wedge \frac{d^{3/4}}{\sqrt{n\alpha^2}}$ | $\leq (n\alpha^2)^{-\frac{2\beta}{4\beta+1}} \wedge \frac{d^{1/4}}{\sqrt{n\alpha^2}}$ $\gtrsim (n\alpha^2)^{-\frac{2\beta+1}{4\beta+3}} \wedge \frac{d^{1/4}}{\sqrt{n\alpha^2}}$ | $(n\alpha^2)^{-\frac{2\beta}{4\beta+2}} \wedge \frac{d^{1/2}}{\sqrt{n\alpha^2}}$ | $\frac{1}{\sqrt{n\alpha^2}}$ |
| $\propto j^\eta e^{-cj^\beta}$ | $\frac{\log^{3/(4\beta)}(n\alpha^2) \wedge d^{3/4}}{\sqrt{n\alpha^2}}$ | $\frac{\log^{1/(4\beta)}(n\alpha^2) \wedge d^{1/4}}{\sqrt{n\alpha^2}}$ | $\frac{\log^{2/(4\beta)}(n\alpha^2) \wedge d^{1/2}}{\sqrt{n\alpha^2}}$ | $\frac{1}{\sqrt{n\alpha^2}}$ |

**Broader impact** In many domains of application, the collection and use of personal data are activities that can have damaging consequences. Data breaches can cause, on the one hand, major distress for the individuals concerned through identity theft and the publication of private information (such as medical or financial records), and, on the other hand, can lead to financial ruin and legal battles for the organizations that are hacked. The study and development of statistical methodology that respects the privacy of individuals has, therefore, the potential for huge impact on society. As the performance of the available methodolgy improves, the need for analysts to use outdated and unsafe procedures will decrease, leading to a positive impact on the world. However, the existence of information theoretic lower bounds proving that private procedures necessarily have worse performance than their non-private counterparts could slightly discourage the use of private methodology, on the grounds of relative inefficiency. Overall, though, this seems like a small price to pay, and a fuller understanding of the possibilites and limitations of private data analysis should be very positive. In this paper we improve upon existing methodology for locally private goodness-of-fit testing, and provide deeper knowledge on the underlying theory.

## Acknowledgments and Disclosure of Funding

T. Berrett and C. Butucea acknowledge financial support from the French National Research Agency (ANR) under the grant Labex Ecodec (ANR-11-LABEX-0047)

T. Berrett acknowledges financial support from the French National Research Agency (ANR) under the grant ANR-17-CE40-0003 HIDITSA

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
