[Supplementary Material]

# Supplementary material to 'Locally private non-asymptotic testing of discrete distributions is faster using interactive mechanisms'

**Berrett Thomas**
CREST, ENSAE, IP Paris
5, avenue Henry le Chatelier
91120 Palaiseau Cedex, FRANCE
`thomas.berrett@ensae.fr`

**Butucea Cristina**
CREST, ENSAE, IP Paris
5, avenue Henry le Chatelier
91120 Palaiseau Cedex, FRANCE
`cristina.butucea@ensae.fr`

## Appendix

### A.1  Proofs of main theorems

*Proof of Theorem 1.*  We first calculate means and variances of our two test statistics, starting with the $U$-statistic $S_B$. Define the function $h : \mathbb{R}^B \times \mathbb{R}^B \to \mathbb{R}$ by

$$h(z_1, z_2) = \sum_{j \in B} \{z_{1j} - p_0(j)\}\{z_{2j} - p_0(j)\}$$

so that $S_B = \frac{1}{n(n-1)} \sum_{i_1 \neq i_2} h(Z_{i_1}, Z_{i_2})$. It is clear that

$$\mathbb{E}S_B = \sum_{j \in B} \{p(j) - p_0(j)\}^2.$$

Now, define

$$\zeta_1 := \mathrm{Var}\big(\mathbb{E}\{h(Z_1, Z_2)|Z_1\}\big) \quad \text{and} \quad \zeta_2 := \mathrm{Var}\big(h(Z_1, Z_2)\big).$$

Using Serfling [1980, Lemma A, p.183] and the fact that $\mathrm{Cov}(Z_{1j}, Z_{1j'}) = \mathbb{1}_{\{j=j'\}}\{p(j)+8/\alpha^2\} - p(j)p(j')$, we have that

$$\binom{n}{2}\mathrm{Var}\, S_B = \sum_{c=1}^{2} \binom{2}{c}\binom{n-2}{2-c}\zeta_c = (2n-3)\zeta_1 + (\zeta_2 - \zeta_1)$$

$$= (2n-3)\mathrm{Var}\bigg(\sum_{j \in B}\{p(j) - p_0(j)\}\{Z_{2j} - p_0(j)\}\bigg)$$

$$+ \mathbb{E}\bigg\{\mathrm{Var}\bigg(\sum_{j \in B}\{Z_{1j} - p_0(j)\}\{Z_{2j} - p_0(j)\} \,\Big|\, Z_1\bigg)\bigg\}$$

$$= 2(n-1)\sum_{j,j' \in B}\{p(j) - p_0(j)\}\{p(j') - p_0(j')\}\mathrm{Cov}(Z_{1j}, Z_{1j'}) + \sum_{j,j' \in B}\mathrm{Cov}(Z_{1j}, Z_{1j'})^2$$

$$= 2(n-1)\sum_{j \in B}\{p(j) + 8/\alpha^2\}\{p(j) - p_0(j)\}^2 - 2(n-1)\bigg(\sum_{j \in B}p(j)\{p(j) - p_0(j)\}\bigg)^2$$

$$+ \sum_{j \in B}p(j)^2\{1 - 2p(j)\} + \bigg(\sum_{j \in B}p(j)\bigg)^2 + \frac{64}{\alpha^4}|B| + \frac{16}{\alpha^2}\sum_{j \in B}p(j)\{1 - p(j)\}$$

$$\leq \frac{18(n-1)}{\alpha^2}\sum_{j \in B}\{p(j) - p_0(j)\}^2 + \frac{82|B|}{\alpha^4}.$$

As a result,
$$\operatorname{Var} S_B \leq \frac{36}{n\alpha^2} \sum_{j \in B} \{p(j) - p_0(j)\}^2 + \frac{164|B|}{n(n-1)\alpha^4}.$$

We now turn to the test statistic $T_B$. First, it is clear that
$$\mathbb{E} T_B = p(B^c) - p_0(B^c).$$

Moreover,
$$\operatorname{Var} T_B = \frac{1}{n}\left(\operatorname{Var} \mathbb{1}_{\{X_{n+1} \in B^c\}} + \frac{4}{\alpha^2}\operatorname{Var} W_{n+1}\right) = \frac{1}{n}\left[p(B)\{1 - p(B)\} + \frac{8}{\alpha^2}\right] \leq \frac{9}{n\alpha^2}.$$

Now, under $H_0$ we have that
$$\mathbb{P}(\phi_B = 1) \leq \mathbb{P}(S_B \geq C_{1,B}) + \mathbb{P}(T_B \geq C_{2,B})$$
$$\leq \frac{n(n-1)\alpha^4\gamma}{656|B|} \times \frac{164|B|}{n(n-1)\alpha^4} + \frac{n\alpha^2\gamma}{36} \times \frac{9}{n\alpha^2} = \frac{\gamma}{2}.$$

Now suppose that we have
$$\delta \geq 8 \max\left[12\left\{\frac{|B|^3}{n(n-1)\alpha^4\gamma^2}\right\}^{1/4}, p_0(B^c)\right], \tag{3}$$

which implies
$$\delta \geq 2 \max\left[24\left\{\frac{|B|^3}{n(n-1)\alpha^4\gamma^2}\right\}^{1/4}, 2p_0(B^c) + \frac{6 + 3\sqrt{2}}{(n\alpha^2\gamma)^{1/2}}\right].$$

Then, under $H_1(\delta, \mathbb{L}_1)$, at least one of
$$\sum_{j \in B} |p(j) - p_0(j)| \geq 24\left\{\frac{|B|^3}{n(n-1)\alpha^4\gamma^2}\right\}^{1/4} \tag{4}$$

or
$$\sum_{j \in B^c} |p(j) - p_0(j)| \geq 2p_0(B^c) + \frac{6 + 3\sqrt{2}}{(n\alpha^2\gamma)^{1/2}} \tag{5}$$

must hold. If (4) holds then we have that
$$\mathbb{P}(S_B < C_{1,B}) \leq \frac{\operatorname{Var} S_B}{[\mathbb{E} S_B - C_{1,B}]^2} \leq \frac{\frac{36}{n\alpha^2}\sum_{j \in B}\{p(j) - p_0(j)\}^2 + \frac{164|B|}{n(n-1)\alpha^4}}{[\sum_{j \in B}\{p(j) - p_0(j)\}^2 - \{\frac{656|B|}{n(n-1)\alpha^4\gamma}\}^{1/2}]^2}$$
$$\leq \frac{\frac{36}{n\alpha^2}\sum_{j \in B}\{p(j) - p_0(j)\}^2}{[\sum_{j \in B}\{p(j) - p_0(j)\}^2 - \{\frac{656|B|}{n(n-1)\alpha^4\gamma}\}^{1/2}]^2} + \frac{\frac{164|B|}{n(n-1)\alpha^4}}{[576\{\frac{|B|}{n(n-1)\alpha^4\gamma}\}^{1/2} - \{\frac{656|B|}{n(n-1)\alpha^4\gamma}\}^{1/2}]^2}$$
$$\leq \frac{144}{n\alpha^2\sum_{j \in B}\{p(j) - p_0(j)\}^2} + \frac{756\gamma}{576^2} \leq \frac{144\gamma}{576} + \frac{756\gamma}{576^2} < \frac{\gamma}{2}.$$

On the other hand, if (5) holds then we have that $\mathbb{E} T_B = p(B^c) - p_0(B^c) \geq \frac{6 + 3\sqrt{2}}{(n\alpha^2\gamma)^{1/2}}$ and hence
$$\mathbb{P}(T_B < C_{2,B}) \leq \frac{\operatorname{Var} T_B}{\{\mathbb{E} T_B - \frac{6}{(n\alpha^2\gamma)^{1/2}}\}^2} \leq \frac{n\alpha^2\gamma}{18} \times \frac{9}{n\alpha^2} = \frac{\gamma}{2}.$$

In conclusion, whenever $H_1(\delta, \mathbb{L}_1)$ holds and $\delta$ satisfies the lower bound in (3), we have that $\mathbb{P}(\phi_B = 0) \leq \gamma/2$, and the result follows.

Under $H_1(\delta, \mathbb{L}_2)$ and using $\sqrt{a + b} \leq \sqrt{a} + \sqrt{b}$:
$$\left(\sum_{j \in B} |p(j) - p_0(j)|^2\right)^{1/2} + \sum_{j \in B^c} |p(j) - p_0(j)| \geq \|p - p_0\|_2 \geq \delta.$$

Now, we suppose that we have instead of (3):

$$\delta \geq 8 \max\left[12\left\{\frac{|B|}{n(n-1)\alpha^4\gamma^2}\right\}^{1/4}, p_0(B^c)\right]$$

$$\geq 2 \max\left[24\left\{\frac{|B|}{n(n-1)\alpha^4\gamma^2}\right\}^{1/4}, 2p_0(B^c) + \frac{6+3\sqrt{2}}{(n\alpha^2\gamma)^{1/2}}\right].$$

That implies, at least one of

$$\left(\sum_{j\in B} |p(j) - p_0(j)|^2\right)^{1/2} \geq 24\left\{\frac{|B|}{n(n-1)\alpha^4\gamma^2}\right\}^{1/4}$$

or

$$\sum_{j\in B^c} |p(j) - p_0(j)| \geq 2p_0(B^c) + \frac{6+3\sqrt{2}}{(n\alpha^2\gamma)^{1/2}}$$

must hold. We conclude similarly the upper bounds for the $\mathbb{L}_2$ test. $\qquad\square$

The proof of Theorem 3 will make use of the following inequality.

**Lemma 1.** *Let $Z \sim N(0,1)$ and $\mu, \lambda > 0$. Then, writing $[x]^\lambda_{-\lambda} = \max\{-\lambda, \min(x, \lambda)\}$, we have that*

$$\mathbb{E}\{[\mu + Z]^\lambda_{-\lambda}\} \geq \frac{1}{2}\min(\mu, \lambda)\min(1, \lambda).$$

*Proof of Lemma 1.* Define

$$h(\mu, \lambda) := \frac{\mathbb{E}\{[\mu + Z]^\lambda_{-\lambda}\}}{\min(\mu, \lambda)} = \frac{\mu + (\lambda - \mu)\bar{\Phi}(\lambda - \mu) - (\lambda + \mu)\bar{\Phi}(\lambda + \mu) - \phi(\lambda - \mu) + \phi(\lambda + \mu)}{\min(\mu, \lambda)}.$$

We now show that, for a fixed $\lambda > 0$, we minimise $h$ by taking $\mu = \lambda$. Indeed, for $\mu > \lambda$ we have

$$\frac{\partial}{\partial\mu}h(\mu, \lambda) = \frac{1}{\lambda}\left\{1 - \bar{\Phi}(\lambda - \mu) - \bar{\Phi}(\lambda + \mu)\right\} > 0.$$

On the other hand, when $\mu < \lambda$ we have

$$\frac{\partial}{\partial\mu}h(\mu, \lambda) = -\frac{1}{\mu^2}\left\{\lambda\bar{\Phi}(\lambda - \mu) - \lambda\bar{\Phi}(\lambda + \mu) - \phi(\lambda - \mu) + \phi(\lambda + \mu)\right\}.$$

Moreover,

$$\frac{\partial}{\partial\mu}\left\{\lambda\bar{\Phi}(\lambda - \mu) - \lambda\bar{\Phi}(\lambda + \mu) - \phi(\lambda - \mu) + \phi(\lambda + \mu)\right\}$$

$$= \mu\left\{\phi(\lambda - \mu) - \phi(\lambda + \mu)\right\} > 0$$

and, as $\mu \searrow 0$, we have

$$\lambda\bar{\Phi}(\lambda - \mu) - \lambda\bar{\Phi}(\lambda + \mu) - \phi(\lambda - \mu) + \phi(\lambda + \mu) = \frac{2}{3}\lambda\mu^3\phi(\lambda) + o(\mu^4) > 0.$$

It therefore follows that when $\mu < \lambda$ we have $\frac{\partial h}{\partial\mu} < 0$. We have now shown that

$$h(\mu, \lambda) \geq h(\lambda, \lambda) = 1 - 2\bar{\Phi}(2\lambda) - \frac{1}{\lambda\sqrt{2\pi}} + \frac{1}{\lambda}\phi(2\lambda).$$

We can check (e.g. numerically) that $h(\lambda, \lambda) \geq \min(1, \lambda)/2$, and the result follows. $\qquad\square$

*Proof of Theorem 3.* Recalling that $\tau = (n\alpha^2)^{-1/2}$, we first consider the expectation of our test statistic $D_B$. Writing $\epsilon_j = \widehat{p}_j - p(j)$, $\Delta_j = p(j) - p_0(j)$ and $\sigma_j^2 = \text{Var}\,\epsilon_j \leq 9/(n\alpha^2) = 9\tau^2$, and letting $Z \sim N(0,1)$, we have

$$\left|\mathbb{E}\{[\widehat{p}_j - p_0(j)]^\tau_{-\tau}\} - \mathbb{E}\{[\Delta_j - \sigma_j Z]^\tau_{-\tau}\}\right|$$

$$= \left|\int_{-\Delta_j}^{\tau - \Delta_j}\{\mathbb{P}(\epsilon_j \geq x) - \mathbb{P}(\sigma_j Z \geq x)\}\,dx - \int_{\Delta_j}^{\tau + \Delta_j}\{\mathbb{P}(\epsilon_j \leq -x) - \mathbb{P}(\sigma_j Z \leq -x)\}\,dx\right|$$

$$\leq 2\tau \sup_{x\in\mathbb{R}}\left|\mathbb{P}(\epsilon_j \leq x) - \mathbb{P}(\sigma_j Z \leq x)\right|$$

$$\leq \frac{C\tau}{\sqrt{n}}\frac{\mathbb{E}\{|\mathbb{1}_{\{X_1=j\}} - p(j) + (2/\alpha)W_{11}|^3\}}{\{p(j)(1 - p(j)) + 8/\alpha^2\}^{3/2}} \lesssim \frac{\tau}{\sqrt{n}},$$

where the final line follows from an application of the Berry–Esseen theorem. Applying this bound and Lemma 1, we therefore have for some universal constant $C > 0$ that

$$
\begin{aligned}
\mathbb{E}D_B &= \sum_{j=1}^{d} \Delta_j \mathbb{E}\big\{[\widehat{p}_j - p_0(j)]_{-\tau}^{\tau}\big\} \geq \sum_{j=1}^{d} \Delta_j \mathbb{E}\big\{[\Delta_j + \sigma_j Z]_{-\tau}^{\tau}\big\} - \frac{C\tau}{\sqrt{n}}\|\Delta\|_1 \\
&\geq \sum_{j=1}^{d} |\Delta_j| \sigma_j \mathbb{E}\big\{[|\Delta_j|/\sigma_j + Z]_{-\tau/\sigma_j}^{\tau/\sigma_j}\big\} - \frac{C\tau}{\sqrt{n}} \\
&\geq \frac{1}{2} \sum_{j=1}^{d} |\Delta_j| \min(|\Delta_j|, \tau) \min(1, \tau/\sigma_j) - \frac{C\tau}{\sqrt{n}} \\
&\geq \frac{1}{6} \sum_{j=1}^{d} |\Delta_j| \min(|\Delta_j|, \tau) - \frac{C\tau}{\sqrt{n}} = \frac{1}{6} D_\tau(p) - \frac{C\tau}{\sqrt{n}},
\end{aligned}
\tag{6}
$$

where we write $D_\tau(p) := \sum_{j=1}^{d} |p(j) - p_0(j)| \min(\tau, |p(j) - p_0(j)|)$. Moreover, under $H_0$ we have that $\mathbb{E}D_B = 0$.

We now turn to the variance of $D_B$. Since the function $x \mapsto [x]_{-\tau}^{\tau}$ is Lipschitz, we have that

$$
\begin{aligned}
\mathrm{Var}\big([\widehat{p}_j - p_0(j)]_{-\tau}^{\tau}\big) &\leq \mathbb{E}\Big\{ \big([\widehat{p}_j - p_0(j)]_{-\tau}^{\tau} - [p(j) - p_0(j)]_{-\tau}^{\tau}\big)^2 \Big\} \\
&\leq \mathrm{Var}(\widehat{p}_j) \leq \frac{1}{n} + \frac{8}{n\alpha^2} \leq \frac{9}{n\alpha^2}.
\end{aligned}
\tag{7}
$$

On the other hand, when $|p(j) - p_0(j)|$ is large, we can prove a tighter bound. Indeed, using a Chernoff bound we have

$$
\begin{aligned}
\mathbb{P}(\widehat{p}_j - p(j) \geq v) &\leq \exp\Big(-\frac{n\alpha^2}{16}v^2\Big) \mathbb{E}[e^{\frac{n\alpha^2 v}{16}\{\widehat{p}_j - p(j)\}}] \\
&\leq \exp\Big(-\frac{n\alpha^2}{16}v^2 + \frac{1}{8n}\Big(\frac{n\alpha^2 v}{16}\Big)^2 - n\log\Big(1 - \frac{4}{n^2\alpha^2}\Big(\frac{n\alpha^2 v}{16}\Big)^2\Big)\Big) \\
&\leq \exp\Big(-\frac{n\alpha^2}{32}v^2\Big).
\end{aligned}
$$

With a similar bound for the lower tail, we thus establish that

$$
|\widehat{p}(j) - p(j)| \leq v, \quad \text{with probability larger than } 1 - 2\exp\Big(-\frac{n\alpha^2 v^2}{32}\Big).
\tag{8}
$$

Thus, when $p(j) - p_0(j) \geq 2\tau$, we have

$$
\begin{aligned}
\mathrm{Var}\big([\widehat{p}_j - p_0(j)]_{-\tau}^{\tau}\big) &\leq \mathbb{E}\big\{ \big(\tau - [\widehat{p}_j - p_0(j)]_{-\tau}^{\tau}\big)^2 \big\} \leq 4\tau^2 \mathbb{P}(\widehat{p}_j - p_0(j) \leq \tau) \\
&\leq 8\tau^2 \exp\Big(-\frac{n\alpha^2}{32}\{p(j) - p_0(j) - \tau\}^2\Big) \leq \frac{8}{n\alpha^2} \exp\Big(-\frac{n\alpha^2\{p(j) - p_0(j)\}^2}{128}\Big),
\end{aligned}
\tag{9}
$$

and we can similarly prove the same bound when $p(j) - p_0(j) \leq -2\tau$. Using (7) and (9), we can see that, for any value of $p(j) - p_0(j)$, we have

$$
\mathrm{Var}\big([\widehat{p}_j - p_0(j)]_{-\tau}^{\tau}\big) \leq \frac{8}{n\alpha^2} \exp\Big(-\frac{n\alpha^2\{p(j) - p_0(j)\}^2}{128}\Big).
\tag{10}
$$

For $j \in [d]$ we will write $P_j := [\widehat{p}_j - p_0(j)]_{-\tau}^{\tau}$ and, for $i \in [n+1]$ and $j' \in [d]$ we will write $\mathbb{E}_i(\cdot) := \mathbb{E}(\cdot|X_1, \ldots, X_{i-1})$ and $\mathbb{E}_i^j(\cdot) := \mathbb{E}(\cdot \times \mathbb{1}_{\{X_i=j\}}|X_1, \ldots, X_{i-1})/p(j)$ for conditional expectations. We will use the fact that $\mathbb{E}_i^{j_1}(P_j) = \mathbb{E}_i^{j_2}(P_j)$ almost surely for any $j_1, j_2 \neq j$ and $i \in [n+1]$. For

$j_1, j_2 \in [d]$ such that $j_1 \neq j_2$, we now consider

$$\mathrm{Cov}\big(P_{j_1}, P_{j_2}\big) = \mathrm{Cov}\Big(\mathbb{E}_{n+1}\big(P_{j_1}\big), \mathbb{E}_{n+1}\big(P_{j_2}\big)\Big)$$

$$= \sum_{i=1}^n \mathbb{E}\big\{\mathbb{E}_{i+1}\big(P_{j_1}\big)\mathbb{E}_{i+1}\big(P_{j_2}\big) - \mathbb{E}_i\big(P_{j_1}\big)\mathbb{E}_i\big(P_{j_2}\big)\big\}$$

$$= \sum_{i=1}^n \mathbb{E}\Big[p(j_1)\mathbb{E}_i^{j_1}(P_{j_1})\mathbb{E}_i^{j_1}(P_{j_2}) + p(j_2)\mathbb{E}_i^{j_2}(P_{j_1})\mathbb{E}_i^{j_2}(P_{j_2}) + \{1 - p(j_1) - p(j_2)\}\mathbb{E}_i^{j_2}(P_{j_1})\mathbb{E}_i^{j_1}(P_{j_2})$$

$$- \big\{p(j_1)\mathbb{E}_i^{j_1}(P_{j_1}) + (1 - p(j_1))\mathbb{E}_i^{j_2}(P_{j_1})\big\}\big\{p(j_2)\mathbb{E}_i^{j_2}(P_{j_2}) + (1 - p(j_2))\mathbb{E}_i^{j_1}(P_{j_2})\big\}\Big]$$

$$= -\sum_{i=1}^n p(j_1)p(j_2)\mathbb{E}\Big[\big\{\mathbb{E}_i^{j_1}(P_{j_1}) - \mathbb{E}_i^{j_2}(P_{j_1})\big\}\big\{\mathbb{E}_i^{j_2}(P_{j_2}) - \mathbb{E}_i^{j_1}(P_{j_2})\big\}\Big]$$

$$= -np(j_1)p(j_2)\mathbb{E}\big[\big\{[n^{-1} + \widehat{p}_{j_1} - p_0(j_1)]_{-\tau}^\tau - [\widehat{p}_{j_1} - p_0(j_1)]_{-\tau}^\tau\big\}$$
$$\times \big\{[\widehat{p}_{j_2} - p_0(j_2)]_{-\tau}^\tau - [\widehat{p}_{j_2} - p_0(j_2) - n^{-1}]_{-\tau}^\tau\big\} \mid X_1 = j_2\big]. \qquad (11)$$

We can therefore always say that, when $j_1 \neq j_2$, we have

$$|\mathrm{Cov}([\widehat{p}_{j_1} - p_0(j_1)]_{-\tau}^\tau, [\widehat{p}_{j_2} - p_0(j_2)]_{-\tau}^\tau)| \leq p(j_1)p(j_2)/n. \qquad (12)$$

However, as before, tighter bound are available when $\max(|p(j_1) - p_0(j_1)|, |p(j_2) - p_0(j_2)|)$ is large. Indeed, if $j \in [d]$ is such that $|p(j) - p_0(j)| \geq 2(\tau + 1/n)$, then, by (8) we have

$$\mathbb{E}\big[\big\{[\widehat{p}_j - p_0(j)]_{-\tau}^\tau - [\widehat{p}_j - p_0(j) - n^{-1}]_{-\tau}^\tau\big\}^2 \mid X_1 = j\big]$$

$$\leq \frac{1}{n^2}\mathbb{P}\bigg(\frac{1}{n}\sum_{i=2}^n \mathbb{1}_{\{X_1=j\}} + \frac{2}{n\alpha}\sum_{i=1}^n W_{ij} - p_0(j) \leq \tau\bigg)$$

$$\leq \frac{1}{n^2}\mathbb{P}\bigg(\bigg|\frac{1}{n}\sum_{i=2}^n \{\mathbb{1}_{\{X_1=j\}} - p(j)\} + \frac{2}{n\alpha}\sum_{i=1}^n W_{ij}\bigg| \geq p(j) - p_0(j) - \tau - \frac{1}{n}\bigg)$$

$$\leq \frac{2}{n^2}\exp\bigg(-\frac{n\alpha^2}{32}\{p(j) - p_0(j) - \tau - 1/n\}^2\bigg) \leq \frac{2}{n^2}\exp\bigg(-\frac{n\alpha^2\{p(j) - p_0(j)\}^2}{128}\bigg). \qquad (13)$$

It now follows from Cauchy–Schwarz, (11), (12) and (13) that, whenever $j_1 \neq j_2$, we have

$$|\mathrm{Cov}([\widehat{p}_{j_1} - p_0(j_1)]_{-\tau}^\tau, [\widehat{p}_{j_2} - p_0(j_2)]_{-\tau}^\tau)|$$

$$\leq \frac{2}{n}p(j_1)p(j_2)\exp\bigg(-\frac{n\alpha^2}{256}\big[\{p(j_1) - p_0(j_1)\}^2 + \{p(j_2) - p_0(j_2)\}^2\big]\bigg). \qquad (14)$$

It now follows from (10), (14) and the fact that $\sup_{x\geq 0} \frac{xe^{-x^2/128}}{x\wedge 1} = 8e^{-1/2}$, that

$$\mathrm{Var}(D_B) = \mathbb{E}\Big\{\mathrm{Var}\big(D_B|Z_1,\ldots,Z_n\big)\Big\} + \mathrm{Var}\Big(\mathbb{E}\big\{D_B|Z_1,\ldots,Z_n\big\}\Big)$$

$$= \frac{c_\alpha^2\tau^2}{n} + \mathrm{Var}\bigg(\sum_{j=1}^d \{p(j) - p_0(j)\}[\widehat{p}_j - p_0(j)]_{-\tau}^\tau\bigg)$$

$$\leq \frac{c_\alpha^2\tau^2}{n} + \frac{8}{n\alpha^2}\sum_{j=1}^d \{p(j) - p_0(j)\}^2\exp\bigg(-\frac{n\alpha^2\{p(j) - p_0(j)\}^2}{128}\bigg)$$

$$+ \frac{2}{n}\bigg\{\sum_{j=1}^d |p(j) - p_0(j)|p(j)\exp\bigg(-\frac{n\alpha^2\{p(j) - p_0(j)\}^2}{256}\bigg)\bigg\}^2$$

$$\leq \frac{c_\alpha^2\tau^2}{n} + \frac{10}{n\alpha^2}\sum_{j=1}^d \{p(j) - p_0(j)\}^2\exp\bigg(-\frac{n\alpha^2\{p(j) - p_0(j)\}^2}{128}\bigg)$$

$$\leq \frac{c_\alpha^2\tau^2}{n} + \frac{80}{n\alpha^2 e^{1/2}}D_\tau(p) \leq \frac{(e+1)^2}{(e-1)^2(n\alpha^2)^2} + \frac{80D_\tau(p)}{e^{1/2}n\alpha^2}. \qquad (15)$$

Under $H_0$, we can now see that

$$\mathbb{P}(D_B \geq C_3) = \mathbb{P}\Big(D_B \geq \frac{e+1}{e-1} \frac{(4/\gamma)^{1/2}}{n\alpha^2}\Big) \leq \frac{\gamma}{4}.$$

As we have already shown in the proof of Theorem 1, we also have that $\mathbb{P}(T_B \geq C_{2,B}) \leq \gamma/4$ under $H_0$, so that the Type I error of our combined test $\psi_B$ is bounded above by $\gamma/2$. Now, suppose that $p$ is such that

$$D_\tau(p) \geq \max\Big\{ (4/\gamma)^{1/2} \frac{4(e+1)}{e-1}, \frac{10240}{e^{1/2}\gamma}, 12C \Big\} \frac{1}{n\alpha^2},$$

where $C$ is the universal constant in (6). For such $p$, it follows from (6) and (15) that

$$\mathbb{P}(D_B < C_3) \leq \frac{\mathrm{Var} D_B}{\{\frac{1}{2} D_\tau(p) - C_3\}^2} \leq \frac{\gamma}{2}.$$

Now, under $H_1(\delta, \mathbb{L}_2)$, we have

$$D_\tau(p) = \sum_{j=1}^d \{p(j) - p_0(j)\}^2 \min(1, \tau/|p(j) - p_0(j)|)$$

$$\geq \min(\|p - p_0\|_2^2, \tau\|p - p_0\|_2) \geq \min(\delta^2, \tau\delta)$$

This proves that

$$\mathcal{E}_{n,\alpha}(p_0, \mathbb{L}_2) \leq \max\Big\{ (4/\gamma)^{1/2} \frac{4(e+1)}{e-1}, \frac{10240}{e^{1/2}\gamma}, 12C \Big\} \frac{1}{(n\alpha^2)^{1/2}}.$$

We now prove the $\mathbb{L}_1$ result. Let $\emptyset \neq B \subseteq [d]$ be given, and suppose that

$$\delta \geq 8 \max\Big[ \Big(\frac{|B|}{n\alpha^2}\Big)^{1/2} \max\Big\{ (4/\gamma)^{1/2} \frac{4(e+1)}{e-1}, \frac{10240}{e^{1/2}\gamma}, 12C \Big\}, p_0(B^c) \Big].$$

Then, under $H_1(\delta, \mathbb{L}_1)$, at least one of

$$\sum_{j \in B} |p(j) - p_0(j)| \geq \Big(\frac{|B|}{n\alpha^2}\Big)^{1/2} \max\Big\{ (4/\gamma)^{1/2} \frac{4(e+1)}{e-1}, \frac{10240}{e^{1/2}\gamma}, 12C \Big\}$$

or

$$\sum_{j \in B^c} |p(j) - p_0(j)| \geq 2p_0(B^c) + \frac{6 + 3\sqrt{2}}{(n\alpha^2\gamma)^{1/2}}$$

holds. If the second of these holds, then, as in the proof of Theorem 1, we have $\mathbb{P}(T_B < C_{2,B}) \leq \gamma/2$. On the other hand, if the first holds, then we have

$$\|p - p_0\|_2^2 \geq \sum_{j \in B} \{p(j) - p_0(j)\}^2 \geq \frac{1}{|B|} \Big(\sum_{j \in B} |p(j) - p_0(j)|\Big)^2$$

$$\geq \max\Big\{ (4/\gamma)^{1/2} \frac{4(e+1)}{e-1}, \frac{10240}{e^{1/2}\gamma}, 12C \Big\}^2 \frac{1}{n\alpha^2},$$

and our interactive test rejects $H_0$ with probability at least $\gamma/2$. Thus,

$$\mathcal{E}^{\mathrm{I}}_{n,\alpha}(p_0, \mathbb{L}_1) \leq 8 \max\Big[ \Big(\frac{|B|}{n\alpha^2}\Big)^{1/2} \max\Big\{ (4/\gamma)^{1/2} \frac{4(e+1)}{e-1}, \frac{10240}{e^{1/2}\gamma}, 12C \Big\}, p_0(B^c) \Big].$$

$\square$

*Proof of Proposition 4.* The minimax risk for testing is

$$\mathcal{R}_{n,\alpha}(p_0, \delta) \geq \inf_{Q \in \mathcal{Q}_\alpha} \inf_{\phi \in \Phi_Q} \sup_{p_\xi \in H_1(\delta), \xi \in \mathcal{V}} \big\{ \mathbb{E}_{p_0}(\phi) + \mathbb{E}_p(1 - \phi) \big\}$$

$$\geq \inf_{Q \in \mathcal{Q}_\alpha} \inf_{\phi \in \Phi_Q} \big\{ \mathbb{E}_{p_0}(\phi) + E_\xi[\mathbb{E}_{p_\xi}(1 - \phi) \cdot I_{p_\xi \in H_1(\delta)}] \big\},$$

where $E_\xi$ is the average with respect to $\xi$ uniformly distributed over $\mathcal{V}$.

Denote by $QP_0^n$ and $QP_\xi^n$ the likelihood of the sample $Z_1, ..., Z_n$ when the original sample is distributed according to $p_0$ and $p_\xi$, respectively. We write

$$E_\xi[\mathbb{E}_{p_\xi}(1-\phi)\cdot I_{p_\xi \in H_1(\delta)}] = E_\xi\left[\mathbb{E}_{p_0}\frac{QP_\xi^n}{QP_0^n}(1-I_{p_\xi \notin H_1(\delta)})\cdot(1-\phi)\right]$$

$$= \mathbb{E}_{p_0}\left[E_\xi\frac{QP_\xi^n}{QP_0^n}(1-I_{p_\xi \notin H_1(\delta)})\cdot(1-\phi)\right] \geq \mathbb{E}_{p_0}\left[E_\xi\frac{QP_\xi^n}{QP_0^n}(1-\phi)\right] - \gamma_1.$$

Back to the minimax risk

$$\mathcal{R}_{n,\alpha}(p_0,\delta) \geq \inf_{Q\in\mathcal{Q}_\alpha}\inf_{\phi\in\Phi_Q}\mathbb{E}_{p_0}(\phi) + \mathbb{E}_{p_0}\left[E_\xi\frac{QP_\xi^n}{QP_0^n}(1-\phi)\right] - \gamma_1$$

$$\geq \inf_{Q\in\mathcal{Q}_\alpha}(1-\eta)\mathbb{P}_{p_0}\left(E_\xi\frac{QP_\xi^n}{QP_0^n} \geq 1-\eta\right) - \gamma_1$$

$$\geq \inf_{Q\in\mathcal{Q}_\alpha}(1-\eta)\left(1 - \frac{1}{\eta}TV(QP_0^n, E_\xi QP_\xi^n)\right) - \gamma_1,$$

for arbitrary $\eta$ in (0,1). $\qquad\square$

*Proof of Theorem 5.* For general sequentially interactive mechanisms, we use the convexity of the Kullback–Leibler discrepancy and the fact that the Kullback–Leibler discrepancy is bounded above by the $\chi^2$ discrepancy to get

$$KL(QP_0^n, E_\xi QP_\xi^n) \leq E_\xi \int m^0(z)\log\frac{m^\xi(z)}{m^0(z)}dz$$

$$= \sum_{i=1}^n E_\xi\mathbb{E}_{p_0}\left[\int \log\frac{m_i^\xi(z_i|Z_1,...,Z_{i-1})}{m_i^0(z_i|Z_1,...,Z_{i-1})}m_i^0(z_i|Z_1,...,Z_{i-1})dz_i\right]$$

$$\leq \sum_{i=1}^n E_\xi\mathbb{E}_{p_0}\left[\int \frac{(m_i^\xi - m_i^0)^2(z_i|Z_1,...,Z_{i-1})}{m_i^0(z_i|Z_1,...,Z_{i-1})}dz_i\right]$$

$$= \sum_{i=1}^n E_\xi\mathbb{E}_{p_0}\left[(p_\xi - p_0)^\top \int \frac{q_i(z_i|\cdot,Z_1,...,Z_{i-1})q_i(z_i|\cdot,Z_1,...,Z_{i-1})^\top}{m_i^0(z_i|Z_1,...,Z_{i-1})}dz_i(p_\xi - p_0)\right]$$

$$= E_\xi\left[(p_\xi - p_0)^\top\Omega(p_\xi - p_0)\right].$$

In the particular case of noninteractive mechanisms, we have

$$\chi^2(QP_0^n, E_\xi QP_\xi^n) = \mathbb{E}_{p_0}\left[\left(E_\xi\frac{m_1^\xi(Z_1)\cdot...\cdot m_n^\xi(Z_n)}{m_1^0(Z_1)\cdot...\cdot m_n^0(Z_n)}\right)^2\right] - 1$$

$$= \mathbb{E}_{p_0}\left[E_{\xi,\xi'}\left(\frac{m_1^\xi(Z_1)\cdot...\cdot m_n^\xi(Z_n)}{m_1^0(Z_1)\cdot...\cdot m_n^0(Z_n)}\frac{m_1^{\xi'}(Z_1)\cdot...\cdot m_n^{\xi'}(Z_n)}{m_1^0(Z_1)\cdot...\cdot m_n^0(Z_n)}\right)\right] - 1$$

$$= E_{\xi,\xi'}\prod_{i=1}^n \mathbb{E}_{p_0}\left[\left(1 + \frac{m_i^\xi(Z_i) - m_i^0(Z_i)}{m_i^0(Z_i)}\right)\left(1 + \frac{m_i^\xi(Z_i) - m_i^0(Z_i)}{m_i^0(Z_i)}\right)\right] - 1$$

$$= E_{\xi,\xi'}\prod_{i=1}^n\left(1 + \mathbb{E}_{p_0}\left[\frac{m_i^\xi(Z_i) - m_i^0(Z_i)}{m_i^0(Z_i)}\frac{m_i^\xi(Z_i) - m_i^0(Z_i)}{m_i^0(Z_i)}\right]\right) - 1.$$

Indeed, $\mathbb{E}_{p_0}[(m_i^\xi(Z_i) - m_i^0(Z_i))/m_i^0(Z_i)] = 0$. Moreover,

$$\chi^2(QP_0^n, E_\xi QP_\xi^n) \leq E_{\xi,\xi'}\exp\left(\sum_{i=1}^n \mathbb{E}_{p_0}(\frac{m_i^\xi(Z_i)}{m_i^0(Z_i)} - 1)(\frac{m_i^\xi(Z_i)}{m_i^0(Z_i)} - 1)\right) - 1$$

$$\leq E_{\xi,\xi'}\exp\left((p_\xi - p_0)^\top\sum_{i=1}^n \mathbb{E}_{p_0}\left[(\frac{q_i^\xi(Z_i|\cdot)}{m_i^0(Z_i)} - 1)(\frac{q_i^{\xi'}(Z_i|\cdot)^\top}{m_i^0(Z_i)} - 1)\right](p_{\xi'} - p_0)\right) - 1$$

$$\leq E_{\xi,\xi'}\left[\exp\left((p_\xi - p_0)^\top\Omega(p_{\xi'} - p_0)\right)\right] - 1.$$

$\square$

*Proof of Theorem 6.* For $i \in [n]$, write $q_i(j|\cdot)$ for the density of $Z_i|\{X_i = j\}$, and write

$$m_0^i(z) := \sum_{j=1}^{d} q_i(z|j)p_0(j).$$

For $j_* \in [d]$ let $B = \{2, \ldots, j_* + 1\}$, and for $j, j' \in B$ and $i \in [n]$ write

$$\omega_{jj'}^i = \int m_0^i(z)\Big\{\frac{q_i(z|j)}{m_0^i(z)} - 1\Big\}\Big\{\frac{q_i(z|j')}{m_0^i(z)} - 1\Big\}\, dz.$$

For each $i \in [n]$, the matrix $\Omega_i := (\omega_{jj'}^i)_{j,j' \in B}$ is a covariance matrix so it is symmetric and non-negative definite. Writing $\bar{\Omega} := n^{-1}\sum_{i=1}^{n}\Omega_i$, then $\bar{\Omega}$ is also symmetric and non-negative definite and hence has real eigenvalues $0 \le \lambda_1 \le \ldots \le \lambda_{j_*}$ and associated eigenvectors $v_1, \ldots, v_{j_*}$. Since $Q$ is $\alpha$-LDP we have that

$$\text{trace}(\bar{\Omega}) = \frac{1}{n}\sum_{i=1}^{n}\text{trace}(\Omega_i) = \frac{1}{n}\sum_{i=1}^{n}\sum_{j \in B}\int m_0^i(z)\Big\{\frac{q_i(z|j)}{m_0^i(z)} - 1\Big\}^2 dz \le (e^\alpha - 1)^2 j_*.$$

Now if we take $j_0 := \max\{j \in B : \lambda_j \le 2(e^\alpha - 1)^2\}$ we have that $j_0 > j_*/2 - 1$. Indeed, if we had $j_0 \le j_*/2 - 1$ then

$$\sum_{j > j_0}^{j_*}\lambda_j > (j_* - j_0)\cdot 2(e^\alpha - 1)^2 \ge (j_* + 2)(e^\alpha - 1)^2,$$

which is in contradiction with the fact that $\sum_j \lambda_j \le j_*(e^\alpha - 1)^2$.

Given a sequence $\xi = (\xi_1, \ldots, \xi_{j_0}) \in \{-1, 1\}^{j_0}$ define $\delta_\xi^j := \sum_{k=1}^{j_0}\xi_k v_{kj}$ for $j \in B$, define $\delta_\xi^+ := \sum_{j \in B}\delta_\xi^j$ and, given $\epsilon > 0$, define

$$p_\xi(j) := \begin{cases} p_0(j)(1 - \epsilon\delta_\xi^+) + \epsilon\delta_\xi^j & \text{if } j \in B \\ p_0(j)(1 - \epsilon\delta_\xi^+) & \text{otherwise} \end{cases}.$$

Note that we have $\sum_{j=1}^{d} p_\xi(j) = 1$. Write $\Xi_\epsilon \subset \{-1, 1\}^{j_0}$ for the set of all sequences $\xi$ such that $|\delta_\xi^+| \le 1/(2\epsilon)$ and $\max_{j \in B}|\delta_\xi^j| \le p_0(j_* + 1)/(2\epsilon)$. Then, for $\xi \in \Xi_\epsilon$, we have $p_\xi \in \mathcal{P}_d$. Given $\xi \in \Xi_\epsilon$ write

$$m_\xi^i(z) = \sum_{j=1}^{d} q_i(z|j)p_\xi(j) = (1 - \epsilon\delta_\xi^+)m_0^i(z) + \epsilon\sum_{j \in B}\delta_\xi^j q_i(z|j)$$

$$= m_0^i(z)\Big[1 + \epsilon\sum_{j \in B}\delta_\xi^j\Big\{\frac{q_i(z|j)}{m_0^i(z)} - 1\Big\}\Big] = m_0^i(z)\Big[1 + \epsilon\delta_\xi^T\Big\{\frac{q_i(z|\cdot)}{m_0^i(z)} - \mathbf{1}\Big\}\Big]$$

where we write $\mathbf{1} = (1, \ldots, 1) \in \mathbb{R}^{j_*}$ for the constant vector, $q_i(z|\cdot) = (q_i(z|2), \ldots, q_i(z|j_* + 1))$ and $\delta_\xi = (\delta_\xi^2, \ldots, \delta_\xi^{j_*}) = \sum_{k=1}^{j_0}\xi_k v_k$. Let $\eta$ be a uniformly random element of $\Xi_\epsilon$, and define

$$Y = E_\eta\Big[\frac{m_\eta^1(Z_1)\ldots m_\eta^n(Z_n)}{m_0^1(Z_1)\ldots m_0^n(Z_n)}\Big] - 1.$$

Let $\eta'$ be an independent copy of $\eta$, and let $\xi, \xi'$ be two independent sequences of Rademacher random variables. Then, using the facts that $1 + x \leq e^x$ for all $x \in \mathbb{R}$ and $\Xi_\epsilon = -\Xi_\epsilon$, we have

$$
\mathbb{E}_{p_0}(Y^2) = E_{\eta,\eta'}\left[\int \frac{m_\eta^1(z_1) m_{\eta'}^1(z_1) \ldots m_\eta^n(z_n) m_{\eta'}^n(z_n)}{m_0^1(z_1) \ldots m_0^n(z_n)} \, dz_1 \ldots dz_n \right] - 1
$$

$$
= E_{\eta,\eta'}\left\{\left(1 + \epsilon^2 \delta_\eta^T \Omega_1 \delta_{\eta'}\right) \ldots \left(1 + \epsilon^2 \delta_\eta^T \Omega_n \delta_{\eta'}\right)\right\} - 1 \leq E_{\eta,\eta'}\left\{\exp\left(n\epsilon^2 \delta_\eta^T \bar{\Omega} \delta_{\eta'}\right)\right\} - 1
$$

$$
= E_{\eta,\eta'}\left\{\exp\left(n\epsilon^2 \sum_{k=1}^{j_0} \eta_k \eta_k' \lambda_k\right) - 1\right\} = E_{\eta,\eta'}\left\{\sum_{\ell=1}^{\infty} \frac{1}{(2\ell)!}\left(n\epsilon^2 \sum_{k=1}^{j_0} \eta_k \eta_k' \lambda_k\right)^{2\ell}\right\}
$$

$$
\leq \frac{1}{P_\xi(\xi \in \Xi_\epsilon)^2} E_{\xi,\xi'}\left\{\sum_{\ell=1}^{\infty} \frac{1}{(2\ell)!}\left(n\epsilon^2 \sum_{k=1}^{j_0} \xi_k \xi_k' \lambda_k\right)^{2\ell}\right\}
$$

$$
= \frac{1}{P_\xi(\xi \in \Xi_\epsilon)^2} E_{\xi,\xi'}\left\{\exp\left(n\epsilon^2 \sum_{k=1}^{j_0} \xi_k \xi_k' \lambda_k\right) - 1\right\}
$$

$$
\leq \frac{1}{P_\xi(\xi \in \Xi_\epsilon)^2}\left\{\exp\left(\frac{n^2\epsilon^4}{2} \sum_{k=1}^{j_0} \lambda_k^2\right) - 1\right\} \leq \frac{\exp\left(2n^2\epsilon^4(e^\alpha - 1)^4 j_0\right) - 1}{P_\xi(\xi \in \Xi_\epsilon)^2}.
$$

We now study $P_\xi(\xi \in \Xi_\epsilon)$. Note that for each $j \in B$ the random variable $\delta_\xi^j$ is subgaussian with variance proxy $\sum_{k=1}^{j_0} v_{kj}^2 \leq 1$. We therefore have [Boucheron, Lugosi and Massart, 2013, Theorem 11.8]

$$
E_\xi\left\{\max_{j \in B} |\delta_\xi^j|\right\} \leq \{2\log(2j_*)\}^{1/2} \quad \text{and} \quad \mathrm{Var}_\xi\left(\max_{j \in B} |\delta_\xi^j|\right) \leq 8\{2\log(2j_*)\}^{1/2} + 2.
$$

Hence, $P_\xi(\max_{j \in B} |\delta_\xi^j| \geq 2\log^{1/2}(2j_*)) \to 0$ as $d \to \infty$. Now $\delta_\xi^+$ is subgaussian with variance proxy

$$
\sum_{k=1}^{j_0}\left(\sum_{j \in B} v_{kj}\right)^2 = \sum_{k=1}^{j_0}(v_k^T \mathbf{1})^2 \leq \|\mathbf{1}\|^2 \leq j_*.
$$

We may therefore take

$$
\epsilon \asymp \min\left\{\frac{1}{j_*^{1/4}(n\alpha^2)^{1/2}}, \frac{p_0(j_* + 1)}{\log^{1/2}(j_*)}, \frac{1}{j_*^{1/2}}\right\}.
$$

Now

$$
\|p_\xi - p_0\|_1 = \epsilon \sum_{j \in B} |\delta_\xi^j - p_0(j)\delta_\xi^+| + \epsilon \sum_{j \in B^c} p_0(j)|\delta_\xi^+| \geq \epsilon \sum_{j \in B} |\delta_\xi^j| - \epsilon|\delta_\xi^+|.
$$

By the Khintchine inequality we have that

$$
\sum_{j \in B} E_\xi |\delta_\xi^j| = \sum_{j \in B} E_\xi\left|\sum_{k=1}^{j_0} \xi_k v_{kj}\right| \geq \frac{1}{2^{1/2}} \sum_{j \in B}\left(\sum_{k=1}^{j_0} v_{kj}^2\right)^{1/2} \geq \frac{1}{2^{3/2}} \sum_{j \in B} \mathbb{1}_{\{\sum_{k=1}^{j_0} v_{kj}^2 \geq 1/4\}}
$$

$$
\geq \frac{1}{2^{3/2}}\left(\sum_{j \in B}\sum_{k=1}^{j_0} v_{kj}^2 - \frac{j_*}{4}\right) = \frac{j_0 - j_*/4}{2^{3/2}} \geq \frac{j_*}{24\sqrt{2}},
$$

where the final inequality follows from the facts that $j_0 > j_*/2 - 1$ and $j_0 \in \mathbb{N}$. Now

$$
\mathrm{Var}_\xi\left(\sum_{j \in B} |\delta_\xi^j|\right) = \mathrm{Var}_\xi\left(\sum_{j \in B}\left|\sum_{k=1}^{j_0} \xi_k v_{kj}\right|\right) \leq E_\xi\left[\left(\sum_{j \in B}\left|\sum_{k=1}^{j_0} \xi_k v_{kj}\right|\right)^2\right].
$$

Denote by $V = \sum_{j \in B}\left|\sum_{k=1}^{j_0} \xi_k v_{kj}\right|$. We can prove that, for $t > 1$,

$$
\begin{aligned}
P_\xi\left(V \geq t\sqrt{2\log(j^*)}\right) &\leq \sum_{j \in B} P_\xi\left(\left|\sum_{k=1}^{j_0} \xi_k v_{kj}\right| \geq t\sqrt{2\log(j^*)}\right) \\
&\leq j^* \exp\left(-t^2 \log(j^*)\right) \leq \exp(-(t^2 - 1)\log(j^*)).
\end{aligned}
$$

Now, $E_\xi[V^2] = \int_0^\infty 2v P_\xi(V \geq v) dv \leq 2j^* + 2\int_{j^*}^\infty v \exp(-v^2/2 + j^*) dv \lesssim j^*$.

Moreover, $E_\xi|\delta_\xi^+| \leq j_*^{1/2}$. Writing $Z := \frac{\sum_{j \in B} |\delta_\xi^j|}{\sum_{j \in B} \mathbb{E}|\delta_\xi^j|}$ we have $\text{Var}_\xi(Z) \leq 1152$ and hence that

$$1 = E_\xi Z \leq \frac{1}{4} + 4612 P_\xi(1/4 \leq Z < 4612) + \frac{\mathbb{E}(Z^2)}{4612} \leq \frac{1}{2} + 4612 P_\xi(Z \geq 1/4).$$

Thus,

$$P_\xi\left(\|p_\xi - p_0\|_1 \geq \frac{\epsilon j_*}{192\sqrt{2}}\right) \geq P_\xi\left(\sum_{j \in B} |\delta_\xi^j| \geq \frac{j_*}{96\sqrt{2}}\right) - P_\xi\left(|\delta_\xi^+| > \frac{j_*}{192\sqrt{2}}\right)$$

$$\geq \frac{1}{9224} - \frac{192\sqrt{2}}{j_*^{1/2}} \geq \frac{1}{10000}$$

for $j_*$ sufficiently large. Thus

$$\mathcal{E}_{n,\alpha}^{\text{NI}}(p_0, \mathbb{L}_1) \gtrsim \epsilon j_* \gtrsim \min\left\{\frac{j_*^{3/4}}{(n\alpha^2)^{1/2}}, \frac{j_* p_0(j_* + 1)}{\log^{1/2}(2j_*)}, j_*^{1/2}\right\}$$

$$= \min\left\{\frac{j_*^{3/4}}{(n\alpha^2)^{1/2}}, \frac{j_* p_0(j_* + 1)}{\log^{1/2}(2j_*)}\right\},$$

and the result follows.

The proof for the $\mathbb{L}_2$ test follows the same lines. It is sufficient to bound from below $\|p_\xi - p_0\|_2$ with high probability. We have

$$P_\xi\left(\|p_\xi - p_0\|_2^2 \geq \frac{1}{144}\varepsilon^2 j_*\right) \geq P_\xi\left(\varepsilon^2\left\{\sum_{j \in B}(\delta_\xi^j)^2 - 2\delta_\xi^+ \cdot \sum_{j \in B} \delta_\xi^j p_0(j)\right\} \geq \frac{1}{144}\varepsilon^2 j_*\right)$$

$$\geq P_\xi\left(\sum_{j \in B}(\delta_\xi^j)^2 \geq \frac{1}{16}j_*\right) - P_\xi\left(2\delta_\xi^+ \cdot \sum_{j \in B} \delta_\xi^j p_0(j) \geq \frac{1}{18}j_*\right),$$

for $j_*$ large enough. Moreover, $\sum_{j \in B} E_\xi(\delta_\xi^j)^2 = \sum_{j \in B} \sum_k v_{kj}^2 = j_0$ by orthonormality of the eigenvectors $v_j$ and

$$E_\xi\left[\left(\sum_{j \in B}(\delta_\xi^j)^2\right)^2\right] = \left(\sum_{j \in B} \sum_{k=1}^{j_0} v_{kj}^2\right)^2 = j_0^2.$$

Therefore, $P_\xi(\sum_{j \in B}(\delta_\xi^j)^2 \geq 2j_0) \leq 1/4$. Denote by $Z = \sum_{j \in B}(\delta_\xi^j)^2$ We get

$$1 = E_\xi(Z/\mathbb{E}Z) \leq \frac{1}{4} + 2P_\xi(Z \geq \mathbb{E}Z/4) + P_\xi(Z \geq 2\mathbb{E}Z) \leq \frac{1}{2} + 2 \cdot P_\xi(Z \geq j_0/4)$$

meaning that $P_\xi(Z \geq j_*/16) \geq P_\xi(Z \geq j_0/4) \geq 1/4$ (as $j_0 \geq j_*/2 - 1 \geq j_*/4$ for $j_*$ large enough). Also

$$P_\xi\left(2\delta_\xi^+ \cdot \sum_{j \in B} \delta_\xi^j p_0(j) \geq \frac{1}{18}j_*\right) \leq \frac{36}{j_*} E_\xi\left[|\delta_\xi^+ \cdot \sum_{j \in B} \delta_\xi^j p_0(j)|\right]$$

$$\leq \frac{36}{j_*}\left(E_\xi(\delta_\xi^+)^2 \cdot E_\xi(\sum_{j \in B} \delta_\xi^j p_0(j))^2\right)^{1/2}$$

$$\leq \frac{36}{j_*} j_*^{1/2}\left(\sum_k \sum_j v_{kj}^2 p_0(j)\right)^{1/2} \leq \frac{36}{j_*^{1/2}},$$

which is less or equal to 1/5 for $j_*$ large enough. Thus

$$\mathcal{E}_{n,\alpha}^{\mathrm{NI}}(p_0, \mathbb{L}_2) \gtrsim \epsilon\sqrt{j_*} \gtrsim \min\left\{\frac{j_*^{1/4}}{(n\alpha^2)^{1/2}}, \frac{j_*^{1/2}p_0(j_*+1)}{\log^{1/2}(2j_*)}, 1\right\}.$$

$\square$

*Proof of Theorem 8.* Let us first prove the bounds for the $\mathbb{L}_2$ norm. When $\epsilon \in [0, 1 - 1/d]$ we can define the probability vector

$$p = (1 - \epsilon)p_0 + (0, \ldots, 0, \epsilon),$$

which satisfies $\|p - p_0\|_1 = \epsilon\{1 - p_0(d)\} \leq \epsilon$ and

$$\|p - p_0\|_2 = \epsilon\left[\{1 - p_0(d)\}^2 + \sum_{j=1}^{d-1} p_0(j)^2\right]^{1/2} \geq \epsilon(1 - 1/d).$$

Thus, using Theorem 1 of Duchi et al. [2018] and taking $\epsilon \leq \frac{1}{\sqrt{8n\alpha^2}}$, we have that

$$\|M_1 - M_0\|_{\mathrm{TV}} \leq \frac{1}{\sqrt{2}}$$

for any sequentially interactive privacy mechanism that takes $p_0$ to $M_0$ and $p$ to $M_1$. We can therefore establish a lower bound of the order of $(n\alpha^2)^{-1/2}$ for the $\mathbb{L}_2$ testing problem.

**Proof of the lower bounds for the $\mathbb{L}_1$-risk, interactive mechanisms** Fix $j_* \in [d]$ and write $B = \{1, \ldots, j_*\}$. Let $Q$ be a sequentially interactive, $\alpha$-LDP privacy mechanism, and for each $i \in [n], j \in [d]$ and $z_1, \ldots, z_{i-1}, z$, write $q(z|j, z_1, \ldots, z_{i-1})$ for the conditional density of $Z_i$ given $X_i = j, Z_1 = z_1, \ldots, Z_{i-1} = z_{i-1}$. For each $i \in [n]$ and $z_1, \ldots, z_{i-1}$ define the $j_* \times j_*$ matrix $\Omega_i(z_1, \ldots, z_{i-1})$ by

$$\Omega_i(z_1, \ldots, z_{i-1})_{j_1 j_2}$$
$$:= \int \{p_0^T q_i(z|\cdot, z_1, \ldots, z_{i-1})\}\left(\frac{q_i(z|j_1, z_1, \ldots, z_{i-1})}{p_0^T q_i(z|\cdot, z_1, \ldots, z_{i-1})} - 1\right)\left(\frac{q_i(z|j_2, z_1, \ldots, z_{i-1})}{p_0^T q_i(z|\cdot, z_1, \ldots, z_{i-1})} - 1\right)^T dz.$$

Consider the $j_* \times j_*$ non-negative definite matrix

$$\Omega := \mathbb{E}_{p_0}\left[\sum_{i=1}^{n} \Omega_i(Z_1, \ldots, Z_{i-1})\right],$$

and write $\lambda_1 \geq \lambda_2 \geq \ldots \geq \lambda_{j_*} \geq 0$ for its eigenvalues and $v_1, \ldots, v_{j_*}$ for its associated eigenvectors, with $v_d = p_0$ and $\lambda_d = 0$ if $j_* = d$. Given a sequence $\xi = (\xi_1, \ldots, \xi_{j_* \wedge (d-1)}) \in \{-1, 1\}^{j_* \wedge (d-1)}$ define $\delta_\xi^j := \sum_{k=1}^{j_* \wedge (d-1)} \xi_k v_{kj}$ for $j \in B$ and define $\delta_\xi^+ := \sum_{j \in B} \delta_\xi^j$. Further, given $\epsilon > 0$, set

$$p_\xi(j) := \begin{cases} (1 - \epsilon\delta_\xi^+)p_0(j) + \epsilon\delta_\xi^j & \text{if } j \in B \\ (1 - \epsilon\delta_\xi^+)p_0(j) & \text{otherwise} \end{cases}.$$

This sums to zero, and when $\epsilon \lesssim p_0(j_*)/\sqrt{\log(2j_*)}$ and $\xi$ is an i.i.d. Rademacher vector, then $p_\xi$ is also non-negative with high probability. Moreover, for each $i \in [n]$ and $z_1, \ldots, z_i$, we have

$$\left|\frac{(p_\xi - p_0)^T q_i(z_i|\cdot, z_1, \ldots, z_{i-1})}{p_0^T q_i(z_i|\cdot, z_1, \ldots, z_{i-1})}\right| \leq e^{2\alpha}\|p_\xi - p_0\|_1 \leq 2e^{2\alpha}\epsilon \sum_{j \in B}\left|\sum_{k=1}^{j_* \wedge (d-1)} \xi_k v_{kj}\right|,$$

and this is $\lesssim \epsilon j_* \to 0$ with high probability. Given $z_1, \ldots, z_n$ and $\xi$ write

$$m_\xi(z_1, \ldots, z_n) = \prod_{i=1}^{n} p_\xi^T q_i(z_i|\cdot, z_1, \ldots, z_{i-1})$$

for the marginal density of $Z_1, \ldots, Z_n$ when $X_1, \ldots, X_n$ have distribution $p_\xi$, and similary define $m_0$ for the density of $Z_1, \ldots, Z_n$ when $X_1, \ldots, X_n$ have distribution $p_0$. Writing $M_\xi$ for the distribution associated with $m_\xi$ and $\bar{M}$ for the mixture distribution $E_\xi(M_\xi)$, we have that

$$\mathrm{KL}(M_0 \| \bar{M}) \le E_\xi[\mathrm{KL}(M_0 \| M_\xi)] = E_\xi\left[\int m_0(z) \log \frac{m_0(z)}{m_\xi(z)}\, dz\right]$$

$$= -\sum_{i=1}^n E_\xi\left[\int \left(\prod_{i'=1}^i p_0^T q_{i'}(z_{i'} | \cdot, z_1, \ldots, z_{i'-1})\right) \log\left(1 + \frac{(p_\xi - p_0)^T q_i(z_i | \cdot, z_1, \ldots, z_{i-1})}{p_0^T q_i(z_i | \cdot, z_1, \ldots, z_{i-1})}\right) dz_1 \ldots dz_i\right]$$

$$\le \sum_{i=1}^n E_\xi\left[\int \left(\prod_{i'=1}^i p_0^T q_{i'}(z_{i'} | \cdot, z_1, \ldots, z_{i'-1})\right) \left(\frac{(p_\xi - p_0)^T q_i(z_i | \cdot, z_1, \ldots, z_{i-1})}{p_0^T q_i(z_i | \cdot, z_1, \ldots, z_{i-1})}\right)^2 dz_1 \ldots dz_i\right]$$

$$= \epsilon^2 \sum_{i=1}^n E_\xi\left[\sum_{j_1, j_2 \in B} \delta_\xi^{j_1} \mathbb{E}_{p_0}\left\{\Omega_i(Z_1, \ldots, Z_{i-1})_{j_1 j_2}\right\} \delta_\xi^{j_2}\right] = \epsilon^2 \sum_{k_1, k_2=1}^{j_* \wedge (d-1)} E_\xi\left[\xi_{k_1} \xi_{k_2} v_{k_1}^T \Omega v_{k_2}\right]$$

$$= \epsilon^2 \sum_{k=1}^{j_* \wedge (d-1)} \lambda_k = \epsilon^2 \mathrm{tr}(\Omega) \lesssim \epsilon^2 j_* n\alpha^2.$$

Now, as in our earlier, non-interactive, lower bound, we have

$$\|p_\xi - p_0\|_1 = \epsilon \sum_{j \in B} \left|\sum_{k=1}^{j_* \wedge (d-1)} \xi_k v_{kj}\right| \gtrsim_p \epsilon j_*.$$

We can then choose $\epsilon \asymp \min\{(j_* n\alpha^2)^{-1/2}, p_0(j_*)/\log^{1/2}(2j_*)\}$ to prove a lower bound of

$$\epsilon j_* \asymp \min\left\{\left(\frac{j_*}{n\alpha^2}\right)^{1/2}, \frac{p_0(j_*)}{\log^{1/2}(2j_*)}\right\}.$$

$\square$

## A.2  Examples

**Polynomially decreasing distributions.** Suppose that $p_0(j) \propto j^{-1-\beta}$ for some $\beta > 0$. Writing $C = 2(1 - 2^{-\beta})^{-1/(\beta+3/4)}$, when $n\alpha^2 \le (d/C)^{2\beta+3/2}$, consider $j = \lceil C(n\alpha^2)^{1/(2\beta+3/2)}\rceil$. Then, when also $n\alpha^2 \ge 1$, we have that

$$\sum_{\ell=j+1}^d p_0(\ell) = \frac{\sum_{\ell=j+1}^d \ell^{-1-\beta}}{\sum_{\ell=1}^d \ell^{-1-\beta}} \le \frac{\int_j^\infty x^{-1-\beta}\, dx}{\int_1^{d+1} x^{-1-\beta}\, dx} \le \frac{j^{-\beta}}{1 - 2^{-\beta}} = \frac{j^{3/4}}{(n\alpha^2)^{1/2}} \frac{j^{-\beta-3/4}(n\alpha^2)^{1/2}}{1 - 2^{-\beta}}$$

$$\le \frac{j^{3/4}}{(n\alpha^2)^{1/2}} \frac{2^{\beta+3/4}}{C^{\beta+3/4}(1 - 2^{-\beta})} = \frac{j^{3/4}}{(n\alpha^2)^{1/2}}.$$

Thus, when $1 \le n\alpha^2 \le (d/C)^{2\beta+3/2}$ we have that $j_* \le \lceil C(n\alpha^2)^{1/(2\beta+3/2)}\rceil$. On the other hand, if $n\alpha^2 > (d/C)^{2\beta+3/2}$ then we will just say that $j_* \le d$. It follows that

$$\mathcal{E}_{n,\alpha}^{\mathrm{NI}}(p_0, \mathbb{L}_1) \lesssim \frac{j_*^{3/4}}{(n\alpha^2)^{1/2}} \lesssim \min\left\{(n\alpha^2)^{-\frac{2\beta}{4\beta+3}}, \frac{d^{3/4}}{(n\alpha^2)^{1/2}}\right\}.$$

More generally, suppose that $p_0(j) \propto j^{-1-\beta} L(j)$ for some slowly-varying function $L : [1, \infty) \to (0, \infty)$. We recall that $L$ is said to be slowly-varying if and only if $\lim_{x \to \infty} L(tx)/L(x) = 1$ for all $t > 0$, and that Karamata's theorem says that

$$\lim_{x \to \infty} \frac{(\gamma - 1)\int_x^\infty t^{-\gamma} L(t)\, dt}{x^{-\gamma+1} L(x)} = 1$$

for any $\gamma > 1$. Writing $c_d := \sum_{\ell=1}^d \ell^{-1-\beta} L(\ell)$, whenever $j \to \infty$ with $j \ll d$ we have that

$$\sum_{\ell=j+1}^d p_0(\ell) = c_d^{-1} \sum_{\ell=j+1}^\infty \ell^{-1-\beta} L(\ell) - c_d^{-1} \sum_{\ell=d+1}^\infty \ell^{-1-\beta} L(\ell) \sim c_d^{-1} \sum_{\ell=j+1}^\infty \ell^{-1-\beta} L(\ell)$$

$$\sim \frac{j^{-\beta} L(j)}{c_d \beta} = \frac{j p_0(j)}{\beta}.$$

Letting $x_{n\alpha^2} := \inf\{x \geq 1 : L(x) < \frac{x^{3/4+\beta}}{(n\alpha^2)^{1/2}}\}$, we can see that

$$\mathcal{E}_{n,\alpha}^{\mathrm{NI}}(p_0, \mathbb{L}_1) \lesssim \frac{\min(x_{n\alpha^2}, d)^{3/4}}{(n\alpha^2)^{1/2}}.$$

Let us discuss the lower bounds. Writing $c = \frac{\beta^2(2\beta+3/2)^2}{2(1-2^{-\beta})^2}$ and $j = \lfloor \{cn\alpha^2/\log(n\alpha^2)\}^{1/(2\beta+3/2)} \rfloor$, when $\log(n\alpha^2) \geq \log c + (2\beta + 3/2)\log 2$ and $\frac{cn\alpha^2}{\log(n\alpha^2)} \leq d^{2\beta+3/2}$, we have that

$$\frac{jp_0(j)}{\log^{1/2}(2j)} = \frac{j^{-\beta}}{\log^{1/2}(2j)\sum_{\ell=1}^d \ell^{-1-\beta}} \geq \frac{\beta j^{-\beta}}{\log^{1/2}(2j)(1-2^{-\beta})} = \frac{j^{3/4}}{(n\alpha^2)^{1/2}}\frac{\beta}{1-2^{-\beta}}\frac{(n\alpha^2)^{1/2}}{\log^{1/2}(2j)j^{\beta+3/4}}$$

$$\geq \frac{j^{3/4}}{(n\alpha^2)^{1/2}}\frac{\beta(2\beta+3/2)^{1/2}}{c^{1/2}(1-2^{-\beta})}\left\{1 + \frac{\log c + (2\beta+3/2)\log 2}{\log(n\alpha^2)}\right\}^{-1/2} \geq \frac{j^{3/4}}{(n\alpha^2)^{1/2}}.$$

Hence, we have $\ell_* \geq j$. On the other hand, when $\frac{cn\alpha^2}{\log(n\alpha^2)} > d^{2\beta+3/2}$ and $\log(n\alpha^2) > c2^{2\beta+3/2}$, we have

$$\frac{dp_0(d)}{\log^{1/2}(2d)} \geq \frac{d^{3/4}}{(n\alpha^2)^{1/2}}\frac{\beta}{1-2^{-\beta}}\frac{(n\alpha^2)^{1/2}}{d^{3/4+\beta}\log^{1/2}(2d)} \geq \frac{d^{3/4}}{(n\alpha^2)^{1/2}},$$

and so $\ell_* = d$. In either case, then,

$$\mathcal{E}_{n,\alpha}^{\mathrm{NI}}(p_0, \mathbb{L}_1) \gtrsim \frac{\{(n\alpha^2)/\log(n\alpha^2)\}^{(3/4)/(2\beta+3/2)} \wedge d^{3/4}}{(n\alpha^2)^{1/2}}$$

$$= \{n\alpha^2\log^{3/(4\beta)}(n\alpha^2)\}^{-2\beta/(4\beta+3)} \wedge \frac{d^{3/4}}{(n\alpha^2)^{1/2}}.$$

More generally, suppose that $p_0(j) \propto j^{-1-\beta}L(j)$ and recall the definition of $x_{n\alpha^2}$ from Example A.2. Taking $j = \min(\lfloor x_{n\alpha^2/\log(n\alpha^2)}\rfloor, d)$ in Theorem 6, we have that

$$\mathcal{E}_{n,\alpha}^{\mathrm{NI}}(p_0, \mathbb{L}_1) \gtrsim \frac{\min(x_{n\alpha^2/\log(n\alpha^2)}, d)^{3/4}}{(n\alpha^2)^{1/2}},$$

which matches our upper bound up to a log factor.

**Exponentially decreasing distributions.** Suppose that $p_0(j) \propto \exp(-j^\beta)$ for some $\beta > 0$. Writing $C$ for a large constant, if $(\frac{1}{4\beta} + \frac{1}{2})\log(Cn\alpha^2) \leq d^\beta$ then consider $j = \lceil \{\log(Cn\alpha^2)/2 - (1 - 1/(4\beta))\log\log(Cn\alpha^2)\}^{1/\beta}\rceil$. Then

$$\sum_{\ell=j}^d p_0(\ell) \leq \frac{\int_j^\infty \exp(-x^\beta)\,dx}{\int_1^{d+1}\exp(-x^\beta)\,dx} \lesssim j^{1-\beta}e^{-j^\beta} \lesssim \frac{\log^{3/(4\beta)}(Cn\alpha^2)}{\sqrt{Cn\alpha^2}} \lesssim \frac{j^{3/4}}{\sqrt{Cn\alpha^2}},$$

and we can therefore see that $j_* \lesssim \log^{1/\beta}(n\alpha^2)$. As a result,

$$\mathcal{E}_{n,\alpha}^{\mathrm{NI}}(p_0, \mathbb{L}_1) \lesssim \min\left\{\frac{\log^{3/(4\beta)}(n\alpha^2)}{(n\alpha^2)^{1/2}}, \frac{d^{3/4}}{(n\alpha^2)^{1/2}}\right\}.$$

Concerning the lower bounds, write $c$ for a small constant and consider $j = \lfloor \{\log(cn\alpha^2)/2 + \log\log(cn\alpha^2)/(4\beta) - \log\log\log(cn\alpha^2)/2\}^{1/\beta}\rfloor$. If $j \leq d$ then we have

$$\frac{jp_0(j)}{\log^{1/2}(2j)} \gtrsim \frac{\log^{1/\beta}(cn\alpha^2)e^{-j^\beta}}{\log^{1/2}(\log(cn\alpha^2))} \gtrsim \frac{\log^{3/(4\beta)}(cn\alpha^2)}{\sqrt{cn\alpha^2}} \gtrsim \frac{j^{3/4}}{\sqrt{cn\alpha^2}}.$$

If, on the other hand, $j > d$, then

$$\frac{dp_0(d)}{\log^{1/2}(2d)} \gtrsim \frac{d\exp(-d^\beta)}{\log^{1/2}(2d)} \gtrsim \frac{\log^{3/(4\beta)}(cn\alpha^2)}{\sqrt{cn\alpha^2}} \gtrsim \frac{d^{3/4}}{\sqrt{cn\alpha^2}}.$$

In either case, then, we have

$$\mathcal{E}_{n,\alpha}^{\mathrm{NI}}(p_0, \mathbb{L}_1) \gtrsim \min\left\{\frac{\log^{3/(4\beta)}(n\alpha^2)}{\sqrt{n\alpha^2}}, \frac{d^{3/4}}{\sqrt{n\alpha^2}}\right\},$$

and this matches our previous upper bound.