[Reviews · NeurIPS 2020]

Review 1

Summary and Contributions: This paper considers the goodness of fit testing problem over general distributions in the local differential privacy model. More specifically, the authors find separation rates for general discrete distributions under this privacy constraint. The authors improve on previous work in this area by [Gaboardi, Rogers 2017][Sheffet, 2018], the privacy mechanisms and tests of which authors claim are sub-optimal. They construct efficient algorithms and test procedures for both the non-interactive and sequentially interactive privacy mechanisms and show that the separation rates are faster in the interactive case and these are optimal, for the most part. They conclude with some examples of testing distributions. The learning problem they are studying is the goodness of fit testing problem over discrete distributions. Specifically, given a known distribution p_0 and n i.i.d samples from an unknown distribution p, they study the problem of testing whether p and p_0 are \delta-far (under L_1 or L_2 norms) or not, but they only consider randomized tests that take as input private data points generated by an interactive and non-interactive \alpha-LDP privacy mechanism. The quantity \delta is called the uniform separation rate. The goal is to find the optimal separation rate (\delta) for an arbitrary distribution p_0 under privacy constraints. The local privacy model is one in which there is no trusted party that ``sanitizes’’ the data of participants, the privatization of data is done on the individual level. Randomized response is a classical example related to this model. In the non-interactive setting, each individual generates a private view of their own data without communicating with anyone else. In the sequentially-interactive setting, individual i has access to the already generated private views of the previous i-1 individuals in order to generate their own private view.

Strengths: The contribution is novel, in particular it shows how the [Valiant, Valiant 2017] result changes under the \alpha-LDP constraint. In terms of techniques, the interactive procedure in the \chi^2-test in the context of discrete distributions is novel and powerful. The paper is relevant to Neurips, as it makes progress on learning questions in the context of interactive protocols that moreover achieve local privacy; these are very exciting topics in current theoretical directions in learning.

Weaknesses: I would have appreciated some intuition before delving into the formal technicalities of the proofs. What is interesting/different/hard about achieving local differential privacy in, the interactive setting vs non-interactive vs non-private settings, more intuitively?

Correctness: In terms of techniques, the high-level strategy is to split the support of p_0 into a main set and a tail set, and combine a \chi^2 test on the main set with a tail-test on the tail set to achieve the optimal separation rate. This technique is not novel to this paper (as claimed by the authors themselves). The private data used by the tests is generated differently in the non-interactive vs interactive settings. In the non-interactive setting, the privacy mechanisms use Laplace randomization to generate private views of original data. In the interactive setting, the authors use techniques similar to those in Butucea et al to design a privacy mechanism that uses an interactive procedure in the \chi^2-test. Overall, the approach and technique appear sound. I haven't fully verified the details of the proofs.

Clarity: The paper is well-written, modulo the comments above about giving more intuition.

Relation to Prior Work: Yes, prior work is clearly described.

Reproducibility: Yes

Additional Feedback: Detailed comments: - Lines 24-33: The authors should include an explanation of non-interactive vs sequentially interactive privacy mechanisms in words here. - Line 49: “our optimality results show that the separation rates are optimal in most usual cases”. What are the unusual cases here? - Lines 74-76: Not sure if this claim was clearly addressed, why are the previous results sub-optimal? - Below line 106: in the def for LDP, shouldn’t we also mention that x \neq x’? - Lines 12-121: it would help the reader if there were citations for these well-known methods - Table 1: It would help to clearly show which rates were known before and their authors, vs which result from the current paper ------- I have read the response and I am happy with the authors clarification and plan to update the paper.


Review 2

Summary and Contributions: The authors study non-asymptotic minimax separation radii in L_1- and L_2-norm for testing if a random variable follows a certain discrete distribution (simple null hypothesis). This is done under local differential privacy constraints on the tests. After setting the scene, the authors go on introducing a non-interactive privacy-mechanism (based on Laplace randomization) in combination with a testing procedure. They state corresponding upper bounds on the separation radii for both norms which depend on the allowed testing risk, sample size, privacy parameter and tested distribution. Afterwards, the same is done with an interactive privacy mechanism. In that case, the L_2-upper bound does not depend on the tested distribution and the bounds are generally smaller in magnitude than in the non-interactive setting. The following section is concerned with corresponding information theoretical lower bounds on the separation radii - it is important to keep in mind that such lower bounds hold for any choice of privacy mechanism and testing procedure. The lower bounds are initially stated at a very general level and do not immediately match the upper bounds given in the past sections, but for three different classes of possible tested distributions (null hypothesis) the authors show that the upper and lower bounds match up to a log-factor at most. Apparently, such results did not exist in L_2-norm and also in L_1-norm, a lower bound had only been found in the special case of a uniform distribution.

Strengths: To my mind, the paper contains extensive significant and novel contributions. They are relevant from the perspective of mathematical statistics (privacy mechanisms and testing procedures and analysing them, information theoretical considerations) and the current relevance of powerful statistical methodology under privacy considerations is obvious. The mathematical level of this paper is very high and the authors were very diligent in deriving their statements.

Weaknesses: I'm not very familiar with the literature on testing under privacy constraints, but in minimax testing without such constraints, testing composite null hypotheses has become more popular. Do you see a chance to come up with results in a setting with composite null hypotheses and privacy (with finitely supported distributions, H_0 may correspond to some simplex/hyerplane or so)? This is clearly not a weakness, but still I suppose it is the right place where to mention it: As the paper's mathematical level is so high, it seems to me that it could also be appropriate for a journal on mathematical statistics.

Correctness: To my mind, the paper is technically sound and the used methods are appropriate for the paper's purposes.

Clarity: Very much so. Although the content is rather abstract and complicated in nature, the authors managed to compile a very clear and well-structured paper. Moreover, the proofs - as far as I studied them - are mostly relatively detailed and easy to follow. Though, for instance, "using a concentration inequality", line 30-31 of the supplementary material, may be a bit vague.

Relation to Prior Work: Yes, the authors clearly and elaborately point out existing results and their new contributions.

Reproducibility: Yes

Additional Feedback: You briefly explain in lines 146-151 that your results actually extend to discrete distributions (with infinite support) rather than just distributions with finite support as introduced in section 2 and indicated in the Theorems and Corollaries. I wonder why you understate your contributions - or did I miss something?


Review 3

Summary and Contributions: The paper proposes mechanisms for identity testing under local differential privacy.

Strengths: The paper proposes interactive and non-interactive mechanisms for identity testing under local differential privacy. The proposed test statistics are relatively easy to implement. They also specialize the results for nearly-uniform, polynomially decreasing, and exponentially decreasing distributions.

Weaknesses: 1. The novelty of the paper in the context of http://proceedings.mlr.press/v89/acharya19b/acharya19b.pdf is not clear. In that paper, authors propose near-optimal mechanisms for uniformity testing and then write " In fact, if we allow simple preprocessing of user observations before applying locally private mechanisms, a reduction argument due to Goldreich [Gol16] can be used to directly convert identity testing to uniformity testing." 2. In line 136: why is the test called chi-squared statistic? It seems to be counting collisions similar to the above paper and hence isn't it a \ell^2_2 statistic? Furthermore, it is known that in the non-private case,  \ell^2_2 statistic is not optimal when p_0 is non-uniform. Given this, is there an intuition on why this test is good for private testing? Post author rebuttal edit: I thank the authors for answering my questions and clarifying the differences between previous and current work. I am raising my score.

Correctness: I have not verified the proofs.

Clarity: The paper is a bit hard to follow. It would be good to explain the algorithms with text and provide some intuition on why they perform well. Other nit: What do the authors mean by "and/or" in lines 142 and 186?

Relation to Prior Work: No. Please see the weakness section above.

Reproducibility: Yes

Additional Feedback: I was wondering if the results for decreasing distributions be extended to envelope classes defined in https://arxiv.org/pdf/0801.2456.pdf


Review 4

Summary and Contributions: This paper considers the question of goodness-of-fit (composite hypothesis testing) under local privacy, in both the L1 (total variation) and L2 sense, i.e., when the null is "all distributions at L1 (or L2) distance at least delta from the reference). They establish (up to constant) tight upper and lower bounds for the question, as a function of all the parameters (in an "instance-specific" way, i.e., as a function of the reference distribution itself). =========== After reading the author feedback: I am satisfied by the response to the points I read, and the promise to detail more some specific aspects of the literature.

Strengths: The claims are sound (correctness of proofs), the results are strong (optimal bounds) and extensive (bounds in two of the most used metrics, TV and L2; in a refined fashion (instance-specific); with examples (specific families of reference distributions). This fits squarely, in my opinion, in the interests of the NeurIPS community: statistical inference (for a fundamental problem, goodness-of-fit) under an increasingly important constraint (local privacy).

Weaknesses: As elaborated below, the main message seems a bit misleading to me: the results seem to show that, for the L1 (TV) version of the problem, there is *no* separation between (sequentially) interactive and non-interactive mechanisms. [What follows is only about the TV testing part, not the L2) Specifically, the authors define "non-interactive" as "independent" mechanisms, i.e., where users do not share anything; and "interactive" as "sequentially interactive." However, one can refine "non-interactive" (following a standard distinction in communication complexity and information theory) as - non-interactive, private-coin: the users are fully independent - non-interactive, public-coin: the users share, ahead of time, a common random seed (independent of their actual input) As shown in previous work (cited in the paper: Acharya et al. (2019), l.282 and l.286), for testing under LDP the separation is among the first two, and the current work shows that sequentially interactive is *no more powerful than* non-interactive, public-coin: [non-interactive, public-coin] ≪ [non-interactive, private-coin] ≍ [sequentially interactive] where the current work establishes the "≍" part.

Correctness: To the best of my ability to check, claims and proofs are correct.

Clarity: Yes. The paper is clear and easy to read.

Relation to Prior Work: I would suggest the authors, when mentioning Valiant-Valiant'17 (and Diakonikolas-Kane'16), also mention the conference version of the former, which was in 2014 (to clarify the timeline with the latter). In addition, I would recommend citing the following paper of Blais, Canonne, and Gur (2017): https://drops.dagstuhl.de/opus/volltexte/2017/7536/ (journal version in 2019). This paper refines the understanding of the "instance-optimal" measure of VV17, showing that it's not actually tight, and suggests another measure (incomparable, and also not fully tight), and relates that to the simpler notion of effective support size. (In particular, there exist classes of simple reference distributions for which the upper bounds in VV'17 differ by a polynomial factor in the domain size).

Reproducibility: Yes

Additional Feedback: - Cannone -> Canonne in the bibliography (twice) - Citation on line 183 (Inference under information constraints) has a peer-reviewed version which appeared in ICML'19. - "Pan-Private Uniformity Testing" appeared in COLT'20 since, it's worth updating the citation. - It may be worth citing http://proceedings.mlr.press/v125/acharya20a.html, which also considers goodness-of-fit (under, among others, local privacy constraints) and provides tight bounds (in a worse-case sense over all reference distributions) as a function of the length of the random seed shared by users, in a non-interactive setting. (I.e., full tradeoff between non-interactive private-coin and public-coin, following the terminology above)

[Author Response · NeurIPS 2020]

We thank all referees for their careful reading, constructive remarks and globally positive appreciations.

♯ 1 and ♯ 4. We will include a short description of different privacy mechanisms. They can be non-interactive (also
known as private-coin) when the users randomize independently the sample they receive, or interactive in the sense that
some information is shared. We consider a large class of sequentially interactive mechanisms where some information
(e.g. the privatized sample) can be transmitted from one user to all the next. From this point of view, the public-coin is
a particular case of sequentially interactive mechanisms where the shared information is the original seed the first user
employed. Our results imply that sequentially interactive mechanisms sharing more information than the seed (e.g.
sharing the previously privatized samples) cannot improve on public-coin mechanisms, as the reviewer ♯ 4 pointed out.

♯ 1 Line 49: The difference between upper and lower bounds e.g. in the non-interactive setup is that we get
$\sum_{j > j_*} p_0(j)$ in the upper bounds and $\ell_* p_0(\ell_*)$ in the lower bounds. We cannot currently exclude the possibility
of pathological cases where these terms strongly differ. There is a logarithmic difference if $p_0(j) \propto j^{-1}$ for $j \leq d$.

Line 74-76: the papers mentioned up to that point obtained slower rates than ours. Broadly, Gaboardi and Rogers
(2017) uses a standard chi-squared statistic calculated on noisy data, while Sheffet uses a standard randomized re-
sponse mechanism which performs poorly in high dimensions, even when paired with the test of Valiant and Valiant
that is optimal in the non-private case.

Line 106: in case $x = x'$ the ratio is 1 and the constraint is still checked.

Lines 120-121 and Table 1: references will be included.

♯ 2 Testing composite hypotheses is certainly most challenging but beyond the scope of this paper.

We will include a reference for the concentration inequality we use.

Indeed, our results can be stated for discrete distributions with infinite support. We wanted to state the rates in terms
of $d$ in order to compare with existing literature. However, our proofs hold for $j$ in $\mathbb{N}$ instead of $j$ from 1 to $d$.

♯ 3 The reduction due to Goldreich (further developed by Acharya et al., AISTATS 2019) gives a way of transferring
upper bounds from uniformity testing to general identity testing when measuring separation using the $\mathbb{L}_1$ norm. The
results in Goldreich establish upper bounds for the general problem that are within a constant factor of the upper
bounds for uniformity testing, though it is known that such upper bounds are generally suboptimal. The extension of
this reduction by Acharya et al. [arxiv:1905.08302, Appendix D] can provide better upper bounds for non-uniform $p_0$,
though the optimality of this approach was not proved and lower bounds do not follow. We directly provide upper and
lower bounds for both $\mathbb{L}_1$ and $\mathbb{L}_2$ norms that are explicit in their dependence on $p_0$. The lower bounds due to Acharya
et al. (AISTATS 2019) in the uniform case apply to public-coin mechanisms, a very specific type of sequentially-
interactive mechanism, while ours hold more generally. We can expand the corresponding discussion in the paper
(second paragraph of Section 1.2) to more clearly discuss our novelty in a revision.

We agree with the reviewer that our test statistic is an $\ell_2$ statistic and to call it chi-square is an abuse of notation that will
be fixed. Indeed, weighted $\ell_2$ statistics are called chi-square and in the non-private setup particular weights depending
on the distribution under the null, $p_0$, have to be employed. We tried to explain in the reduced space available, lines
160-167, that the variance of the statistic corresponding to an outcome $j$ depends on $p_0(j)$ in the non-private setup -
hence the weights, but it is free of $p_0(j)$ in the private setup (homoscedasticity) and therefore, no weights are required
here.

We will replace some formulas by text in order to explain the algorithms.

Lines 142 and 186: by 'and/or' we mean that it is not an exclusive or (xor), so that both conditions may hold simulta-
neously. It is probably sufficient to keep 'or' instead of 'and/or'.

The envelope classes considered in arxiv:0801.2456 can be used in our upper bound results in order to state uniform
results with respect to $p_0$ belonging to such an envelope class. However, the lower bounds cannot hold uniformly
for such classes, which means that the upper bounds will be suboptimal for many distributions in the envelope class.
For example, the exponentially decreasing distribution belongs to the envelope class with polynomially decreasing
envelope but the optimal rate for testing it is much faster. This question is related to the point made by reviewer ♯ 2
about composite null hypotheses.

♯ 4 Thank you for the details on private vs. public coin privacy mechanisms. We will include a discussion of this in a
revision.

We will include the additional references, correct the typos and update the full references in the final manuscript.

[Meta-Review · NeurIPS 2020]

This paper makes strong technical contributions to the literature on differentially private hypothesis testing. In particular, the paper addresses the problem of goodness-of-fit under local differential privacy, and provides optimal tests and separation rates (under L1 and L2 distances). These results significantly improve over prior work. The paper also gives efficient algorithms and test procedures for both the non-interactive and sequentially interactive privacy mechanisms. All the reviewers agree that the results are strong and the paper is technically rich.